



# The contribution of Humboldt Glacier, North Greenland, to sea-level rise through 2100 constrained by recent observations of speedup and retreat

Trevor R. Hillebrand[1], Matthew J. Hoffman[1], Mauro Perego[2], Stephen F. Price[1], Ian M. Howat[3,4]

[1]Fluid Dynamics and Solid Mechanics Group, Los Alamos National Laboratory, Los Alamos, NM, 87545, USA
[2]Center for Computing Research, Sandia National Laboratories, Albuquerque, NM 87185, USA
[3]Byrd Polar and Climate Research Center, Columbus, OH, 43210, USA
[4]School of Earth Sciences, Ohio State University, Columbus, OH, 43210, USA

*Correspondence to*: Trevor R. Hillebrand (trhille@lanl.gov)

**Abstract.** Humboldt Glacier, North Greenland, has retreated and accelerated through the 21st century, raising concerns that it could be a significant contributor to future sea-level rise. We use a data-constrained ensemble of three-dimensional higher-order ice sheet model simulations to estimate the likely range of sea-level rise from the continued retreat of Humboldt

Glacier. We first solve for basal traction using observed ice thickness, bed topography, and ice surface velocity from the year 2007 in a partial differential equation constrained optimization. Next, we impose calving rates to match mean observed 2007–2017 retreat rates in a transient calibration of the exponent in the power-law basal friction relationship. We find that power law exponents in the range of 1/7–1/5 — rather than the commonly used 1/3–1 — are necessary to reproduce the observed speedup over this period. We then tune an iceberg calving parameterization based on the von Mises stress yield

criterion in another transient calibration step from 2007–2017 to approximate both observed ice velocities and terminus position in 2017. Finally, we use the range of basal friction relationship exponents and calving parameter values to generate the ensemble of model simulations from 2007–2100 under three climate forcing scenarios from CMIP5 (two RCP 8.5 forcings) and CMIP6 (one SSP5-8.5 forcing). Our simulations predict 5.5–9.2 mm of sea-level rise from Humboldt Glacier, significantly higher than a previous estimate (~3.5 mm) and equivalent to a substantial fraction of the 40–140 mm predicted

by ISMIP6 from the whole Greenland Ice Sheet. Our larger future sea-level rise prediction results from the transient calibration of our basal friction law to match the observed 2007–2017 speedup, which requires a semi-plastic bed rheology. In many simulations, our model predicts the growth of a sizable ice shelf in the middle of the 21st century. Thus, atmospheric warming could lead to more retreat than predicted here if increased surface melt promotes hydrofracture of the ice shelf. Our data-constrained simulations of Humboldt Glacier underscore the sensitivity of model predictions of

Greenland outlet glacier response to warming to choices of basal shear stress and iceberg calving parameterizations. Further, transient calibration of these parameterizations, which has not typically been performed, is necessary to reproduce observed behavior. Current estimates of future sea-level rise from the Greenland Ice Sheet could, therefore, contain significant biases.



## 1 Introduction

The Greenland Ice Sheet (GrIS) could contribute up to 30 cm to global mean sea level in the 21st century, but uncertainty in the magnitude of sea level rise (SLR) under a given emissions scenario exceeds 50% (Goelzer et al., 2020; Edwards et al., 2021). Thus, a better understanding of the processes driving past and present retreat is necessary to reduce uncertainties in forecasts of GrIS mass loss. The retreat and acceleration of marine-terminating outlet glaciers has contributed roughly half of the net mass loss from the GrIS since the early 1990s, while increasingly negative surface mass balance has contributed the

remaining half (IMBIE Team, 2019). The retreat of these outlet glaciers from their 20th century extents may be due to increased access of warm Atlantic Water to glacier termini from 1998–2007 (Wood et al., 2021). However, outlet glaciers experienced substantial thinning from 1985–2000, prior to the onset of rapid terminus retreat, with total ice discharge through outlet glaciers increasing rapidly by 14% around the year 2000 (King et al., 2020). This step increase in discharge was sufficient to shift the GrIS into a state of negative mass balance, with the future annual probability of net mass gain

estimated at around 1% (King et al., 2020). While increased solid ice discharge is expected to remain a primary contributor to SLR from the GrIS over the course of this century (Choi et al., 2021), the precise mechanisms responsible for observed retreat and increased discharge remain poorly understood (King et al., 2020). Compounded with the wide range of polar warming predicted by climate models, this limits the precision of model-based estimates of future SLR (Barthel et al., 2020; Goelzer et al., 2020; Edwards et al., 2021; Payne et al., 2021). Thus, a process-based understanding of recent changes at the

scale of individual outlet glaciers and estimates of the future retreat consistent with these changes is necessary in order to make precise and accurate estimates of the future SLR contribution of ice dynamics from the GrIS.

         Despite widespread retreat of outlet glaciers of the northern GrIS, the velocity response has varied significantly from glacier to glacier (Hill et al., 2018). However, mass loss could accelerate in the future as glaciers retreat over beds that deepen inland, leading to a positive feedback known as the marine ice sheet instability that drives unstable retreat

(Weertman, 1974; Schoof, 2007). Humboldt Glacier (Figure 1) is one of the largest of the northern GrIS outlet glaciers and the widest outlet glacier of the GrIS, with a calving front almost 100 km across. It contains enough ice to raise global mean sea level by 19 cm if melted entirely (Rignot et al., 2021). Humboldt has lost the greatest area of grounded ice since 1992 of any GrIS outlet glacier (Box & Decker, 2011; Wood et al., 2021), with retreat concentrated almost entirely along the fast-flowing northern section of its marine terminus (Figure 1; Carr et al., 2015). This northern section of the terminus is

underlain by a deep basal trough that extends >70 km towards the ice sheet interior, sections of which deepen inland, raising concerns that Humboldt Glacier could be entering a phase of rapid, runaway retreat (Carr et al., 2015). The wide calving front and complex bed geometry of Humboldt Glacier make it particularly difficult to predict its future dynamical response to environmental changes. Furthermore, recent marine bathymetric surveys along the glacier terminus have shown that the glacier bed is up to 200 m deeper than previously thought (Rignot et al., 2021), indicating both that Humboldt Glacier is

exposed to more oceanic heat and that previous modeling studies (Carr et al., 2015; Goelzer et al., 2020; Choi et al., 2021) may be hampered by inaccurate bed geometry.



In this paper, we use a three-dimensional, higher-order, thermomechanically coupled ice sheet model (Hoffman et al., 2018) to estimate the SLR contribution from Humboldt Glacier from 2007–2100 and assess sources of uncertainty. From an optimized initial condition, we perform hindcasting simulations from 2007–2017 to tune a number of model parameters against observed changes at Humboldt Glacier. We then use the set of tuned model parameters in an ensemble of twenty-four simulations to the year 2100 to estimate the likely range of SLR from Humboldt Glacier forced by three climate projections based on RCP8.5 and SSP5-8.5 emissions scenarios. Finally, we explore additional sources of uncertainty in a set of sensitivity experiments that examine the impacts of assumptions about iceberg calving, oceanic melt, and bed topography.

**Figure 1: Observations and optimization of Humboldt Glacier for the initial condition in the year 2007.** (a) BedMachine v4 bed topography (Morlighem et al., 2017, 2021) interpolated onto the 1–10 km regional mesh. Cyan contour indicates bed topography at sea level in all panels. Black contour denotes ice edge. 250 m thickness contours are shown in white, with thick contours marking 1000 m intervals. Inset at upper right shows the location of our model domain in light blue, over a hillshade image from Howat et al. (2014). Inset at lower left shows USGS Landsat imagery of the northern section of the marine terminus within the black box (Howat, 2017), with



terminus positions from 2000 to 2017 (Moon & Joughin, 2008; Joughin et al., 2017). (b) Observed 2007–2008 ice surface velocity (Joughin et al., 2010; 2018) on a logarithmic color scale, interpolated onto our regional mesh. The inset at lower left shows velocity transects over time along the approximate flowline marked in blue in the lower left inset of panel a, using the same color scale. (c) Magnitude of basal shear stress for our 2007 optimized initial condition and (d) the resulting modeled ice surface velocities. Panels b, c, and d are trimmed to the 2007 ice extent, while panel a shows the entire domain.

## 2 Methods

### 2.1 Ice sheet model

We model the evolution of Humboldt Glacier over a regional domain on a 1–10km variable resolution mesh (Figure 1; 18544 cells with 10 vertical levels) using the MPAS-Albany Land Ice (MALI) model (Hoffman et al., 2018). MALI is a thermodynamically coupled, hybrid finite element-finite volume code that solves the first-order Stokes approximation for momentum balance in three dimensions (Blatter, 1995; Pattyn, 2003), using Nye's generalization of Glen's flow law as the constitutive law (Glen, 1955; Nye 1957). We impose Dirichlet velocity boundary conditions around the domain edges using the observed 2007–2008 surface velocity from Joughin et al. (2018). This prevents the possibility of catchment boundary migration, but this is a minor error over the time scales considered here. For basal friction, we use an effective pressure-dependent power-law relationship of the form $\tau_b = N \mu |u|^{q-1} u$, where $\tau_b$ is basal shear stress, $N$ is the effective pressure at the glacier bed, $\mu$ is the spatially varying friction parameter, $u$ is the velocity at the glacier bed, and $0<q\leq1$ is a power-law exponent discussed in more detail below (Budd et al., 1979; 1984). N is calculated as ice overburden pressure minus subglacial water pressure with the commonly used simple assumption of perfect connectivity between the subglacial hydrologic system and the ocean (e.g., Vieli et al., 2001; Seroussi et al., 2013; Asay-Davis et al., 2016; Morlighem et al., 2019). For the energy balance, we use an enthalpy formulation, as described by Hoffman et al. (2018) and based on Aschwanden et al. (2012). We use a map of basal heat flux from Shapiro and Riztwoller (2004).

### 2.2 Data sources and optimized initial condition

We use a PDE-constrained, adjoint optimization method to solve for the basal friction parameter $\mu$ to create our initial condition for the year 2007. This procedure minimizes the sum of the modeled velocity misfit to observations and a regularization term (Figure 1; see also Perego et al., 2014) while constraining the ice velocity and temperature to satisfy the coupled first-order Stokes and enthalpy equations. We set q=1/3 for the optimization, but we later tune this value to find the values that best match observed changes in velocity during retreat in a set of forward runs, as described further below. Ice thickness and bed topography are from BedMachine v4 (BMv4; Morlighem et al., 2017; 2021), with a nominal date of 2007. Observed 2007–2008 ice velocities and their uncertainties are taken from Joughin et al. (2018), with areas of missing velocity filled with velocities from BedMachine. While the BedMachine velocities do not correspond to a given year, the data gaps are found only in slow-moving ice far from the marine terminus, and thus this has a negligible effect on our results.



The initial ice extent from BMv4 is trimmed slightly to match the ice margin as defined in the velocity datasets. Surface mass balance, surface air temperature, subglacial runoff, and ocean thermal forcing fields are taken from the Ice Sheet Model Intercomparison Project for CMIP6 (ISMIP6) (Nowicki et al., 2020; Slater et al., 2020), which provides these outputs from a

number of earth system models. We use the MIROC5 RCP8.5, HadGEM2 RCP8.5 products recommended by ISMIP6 to represent medium and strong anthropogenic forcing, respectively (Barthel et al., 2020; Slater et al., 2020; see their Figure 10b). We also include the CNRM-CM6 SSP5-8.5 forcing to represent the mid-range of the few available CMIP6 forcings, which are known to be among those with the highest climate sensitivity in CMIP6 (Payne et al., 2021). Climate forcing time series are shown in Figure 2.


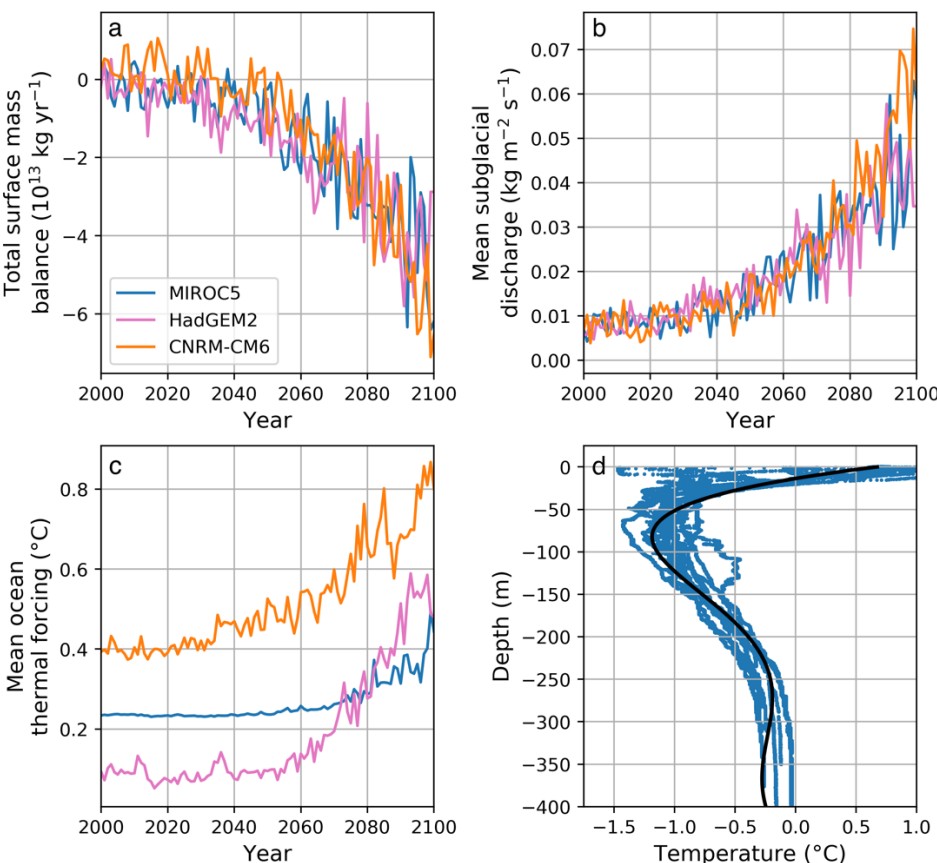

**Figure 2**: a, b, c: Time series of climate forcings from MIROC5, HadGEM2, and CNRM-CM6 earth system models, provided by ISMIP6. These were calculated by summing or averaging over the ice extent in the 2007 initial condition, and so values in these plots do not take into account the retreating ice margin. d: Ocean temperature observations (blue) from Oceans Melting Greenland, used for creating three-
dimensional ocean thermal forcing fields from the two-dimensional fields provided by ISMIP6. Black curve shows polynomial fit.





## 2.3 Tuning of basal friction law exponent

To find the appropriate range of values of m in the basal friction relationship, we recalculate the friction parameter $\mu$ for

$1/10 \leq q \leq 1$ using the relationship $\mu = \mu_{opt}|u_{opt}|^{\frac{1}{3}-q}$, where $\mu_{opt}$ and $u_{opt}$ are the friction parameter and basal sliding speed,

respectively, from the optimization using q=1/3. This relation gives a $\mu$ field that yields the same basal traction and velocity as the optimization. For each value of $q$ considered, we impose a spatially variable calving-front retreat rate that matches the mean observed retreat rate between 2007 and 2018 ice extent from Joughin et al. (2018). The ice thickness and velocity fields are allowed to evolve in response. We calculate the root mean-square error of the modeled velocities from the observed 2017–2018 velocities divided into three bands to determine an appropriate range of values for q (Figure 3). Aside

from the imposed calving rate, the only other forcing in this step is the MIROC5 RCP8.5 surface mass balance, as the imposed calving rate represents the sum of retreat rates due to calving and melt undercutting in reality, the individual components of which are not readily available from observations. We find that $1/7 \leq q \leq 1/5$ yields reasonable agreement to the 2017–2018 velocities in the fastest-flowing parts of the glacier, without causing problematic amounts of acceleration elsewhere. Thus, we use values of 1/5 and 1/7 for q in our forward runs.





Figure 3: a. Maps of modeled and observed velocities used for tuning basal friction law exponent using an imposed retreat rate to match the observed 2017–2018 margin. Modeled velocities are shown in July 2017. Observed (white) and modeled (cyan) velocities are contoured at 100, 300, and 600 m yr[-1]. b. Root mean square (RMS) velocity misfits between model and observation for all basal friction law exponents examined. Dot colors correspond to velocity bands in the observations from 2017–2018.

## 2.4 Submarine melting parameterizations

To represent melt undercutting of the grounded marine terminus, we use the relationship $q_m = \left( A\, h\, q_{sq}^{\alpha} + B \right) TF^{\beta}$ (Rignot et al., 2016; Slater et al., 2020), where $q_m$ is the horizontal melt rate perpendicular to the calving front in m day[-1], A = 0.0003 m$^{\alpha}$ day$^{\alpha-1}$ °C$^{-\beta}$, h is water depth at the terminus in meters, $q_{sq}$ is subglacial runoff in m day[-1], $\alpha$ = 0.39, B = 0.15 m day[-1] °C$^{-\beta}$,





TF is ocean thermal forcing in °C (taken from the ISMIP6 two-dimensional thermal forcings), and $\beta = 1.18$. This melt rate is imposed at grounded marginal cells if there is no floating ice.

In preliminary model runs, we found that the modeled glacier develops a sizable ice shelf by the middle of the 21st century in many simulations. Therefore, we also include a basal melt parameterization for floating ice developed for Antarctic ice shelves (Jourdain et al., 2020). Other studies have used melt formulations in which the relationship between

sub-shelf melt and thermal forcing is linear (Choi et al., 2021) or weakly nonlinear (Cai et al., 2017). However, the currently accepted theoretical formulation is for melt to be a function of thermal forcing squared (Holland et al., 2008), as we apply here. We constructed three dimensional ocean thermal forcing fields using all available CTD and AXCTD measurements from the Oceans Melting Greenland campaign (OMG 2020a,b) within the bounding box [79.0°N, 70.0°W; 79.9°N, 64.0°W] between 2016 and 2020. First, we converted the temperature measurements to thermal forcing using the relationship $TF =$

$T - T_{freeze}$, where $T_{freeze} = aS_{ref} + b + cz$. Here, $a$ = -0.0575 °C PSU$^{-1}$, $S_{ref}$ is a reference salinity of 34.4 PSU, $b$ = 0.0901°C, $c$ = 7.61•10$^{-4}$ °C m$^{-1}$, and $z$ is depth (Jenkins, 1991). We then fit a fifth-order polynomial to these data (Figure 2d) to obtain TF at the top and bottom of the thermocline (80 and 250 m, respectively). We use four vertical ocean levels of depths 0, 80, 250, and 1000 m, for which the top two and bottom two are set to the same respective temperature values, but TF is allowed to vary based on the pressure-dependence of T$_{freeze}$. The model interpolates linearly between these levels based

on ice shelf draft. For each ISMIP6 ocean thermal forcing, this vertical relationship is then translated based on the 2D thermal forcing, taken to be TF at the bottom of the thermocline (250 m depth). We then tune the TF–melt relationship to obtain mean melt rates of 20 m yr$^{-1}$ beneath floating ice in our initial condition. In forward runs, this also produces mean melt rates of ~20 m yr$^{-1}$ over the 2007–2017 period, despite the evolving ice thickness. The mean melt rate of 20 m yr$^{-1}$ was chosen as a representative value for sub-shelf melt near the grounding line of nearby Petermann Glacier (Cai et al., 2017),

and may not be entirely appropriate for Humboldt Glacier; however, we view this as the best estimate available given the lack of ice shelf melt rate estimates at Humboldt Glacier and the considerably smaller amount of floating ice. To investigate the sensitivity of our results to this choice of melt tuning, we include a set of forward simulations with ice shelf melt tuned to 10 m yr$^{-1}$ and 30 m yr$^{-1}$.

**2.5 Calving parameterizations**

Development of significant floating ice is not typically expected in Greenland, and some modeling studies have required ice to calve as soon as it goes afloat (e.g., Morlighem et al., 2016). We find that removing all floating ice leads to too rapid retreat over the observed period, and therefore we do not require floating ice to calve, but instead use two calving parameterizations that each treat floating and grounded ice the same. The first is the von Mises stress calving parameterization of Morlighem et al. (2016): $c = |v| \frac{\sigma}{\sigma_{max}}$ , where $c$ is calving velocity, $v$ is the depth-averaged ice velocity,

$\sigma$ is the depth-averaged tensile von Mises stress, and $\sigma_{max}$ is a yield stress that we treat as a tuning parameter. Thus, the calving rate exceeds the ice velocity for $\sigma > \sigma_{max}$, causing the calving front to retreat. We use the same value of $\sigma_{max}$ for



grounded and floating ice. Choi et al. (2021) found that the threshold stress needed to be as low as 150 kPa to match observed retreat for some Greenland outlet glaciers, which is far below estimates for the tensile strength of ice reported by Petrovic (2003), which range from 700 kPa to 3.1 MPa. We also find that very low values of $\sigma_{max}$ ($\leq$ 200 kPa) are necessary

to cause retreat at Humboldt Glacier. However, we find that this leads to a positive feedback in which increased velocity causes both increased calving and decreased basal friction due to thinning, which in turn causes higher velocities; this is exacerbated by the non-linear basal friction relationship. In preliminary forward runs, this led to unrealistic rates of calving and ice flow after ~2030. We found it necessary to impose a 3 km yr$^{-1}$ upper limit on the calving rate following Choi et al. (2018). While this is an ad hoc solution, it avoids the far more problematic effect of the positive feedback between velocity

and calving rate. We explore the impact of this assumption in a sensitivity experiment described below.

To tune the von Mises tensile threshold stress parameter, $\sigma_{max}$, we ran the model from 2007–2017 with values of $\sigma_{max}$ of 150, 160, 170, 180, 190, and 200 kPa. We compared the fit of the modeled ice margin position and velocity field to observations at 2017–2018 (Joughin et al., 2018). For each climate forcing and basal friction law pair, we chose three values of $\sigma_{max}$ for century-long projection runs. Values of $\sigma_{max}$ that minimized the misfit to observed surface velocities and ice front

positions are in the range of 160–190 kPa (Figure S1; Table 1).

We recognize that uncertainties and biases inherent in the calving parameterization could have a large impact on mass-loss projections. To address this, we also include a set of minimum mass loss simulations, similar to those of Price et al. (2011), which branch off from the basal friction law exponent tuning runs in 2017. After the tuning run with an imposed retreat rate from 2007–2017, we allow only enough calving to hold the glacier terminus at its 2017 position. The terminus

may retreat behind this position due to surface mass balance and ocean thermal forcing, but no calving is active inward of the 2017 terminus. While this is not likely to be an accurate depiction of glacier retreat over the 21st century, we view it as a good estimate of a lower bound on mass loss, as we believe Humboldt Glacier is very unlikely to advance significantly beyond its 2017 position under the climate scenarios examined in this study.

## 2.6 Forward simulations

### 2.6.1 Control

For each pair of basal friction law and climate forcing, we run a control simulation without calving or submarine melting, imposing only the mean 1960–1989 surface mass balance as a forcing, through the entire run from 2007–2100. Ice that advances beyond the 2007 terminus is immediately melted away to avoid advance into regions of unconstrained basal friction coefficient. This serves to quantify the mass loss and equivalent sea-level contribution from ongoing ice flow

adjustment to historical forcing alone. We later subtract this committed mass loss from the perturbed parameter simulations discussed below to isolate mass loss and sea-level rise due to future climate changes.



### 2.6.2 Perturbed parameter ensemble

For each pair of climate forcing and basal friction law, we perform four simulations from 2007–2100 (Table 1): 1) a minimum mass-loss scenario with time-varying surface mass balance and submarine melt forcing, but with only enough calving to keep the glacier from advancing beyond the 2017–2018 margin. These runs branch off of the 2007–2017 basal friction tuning runs described above at year 2017; 2–4) three scenarios with the von Mises calving law, submarine melting, and surface mass balance forcing, using the three values of $\sigma_{max}$ that best match the observed 2007–2017 retreat and speedup. In all runs, ice is again not allowed to advance past its 2007 margin to prevent advance onto areas of unconstrained basal friction, although this only has an effect early in the minimum mass-loss runs. We also include two simulations with q=1 using $\sigma_{max}$ = 150 kPa and 180 kPa for each climate forcing to illustrate the effect of tuning the power-law basal friction relationship; these linear basal friction law simulations should not be taken as estimates of SLR.

Table 1: Runs conducted with the perturbed parameter ensemble, with $\sigma_{max}$ for all members using the von Mises stress parameterization.

|  | MIROC5 (kPa) | HadGEM2 (kPa) | CNRM-CM6 (kPa) |
| --- | --- | --- | --- |
| **q=1/5** | | | |
| 2017 calving front | N/A | N/A | N/A |
| high $\sigma_{max}$ | 180 | 180 | 180 |
| medium $\sigma_{max}$ | 170 | 170 | 170 |
| low $\sigma_{max}$ | 160 | 160 | 160 |
| **q=1/7** | | | |
| 2017 calving front | N/A | N/A | N/A |
| high $\sigma_{max}$ | 180 | 190 | 190 |
| medium $\sigma_{max}$ | 170 | 180 | 180 |
| low $\sigma_{max}$ | 160 | 170 | 170 |

### 2.6.3 Sensitivity experiments

In addition to the main perturbed parameter ensemble, we include sensitivity experiments to explore the role of floating ice, maximum calving rates, and bed topography in the evolution of Humboldt Glacier. All sensitivity experiments use the intermediate value of $\sigma_{max}$ for the appropriate climate forcing–basal friction exponent pair.

In the first sensitivity experiment, we remove all floating ice starting in 2055 and thereafter require ice to calve as soon as it goes afloat, while keeping the von Mises calving law the same for grounded ice. We choose 2055 as the branch point



because this is approximately when the surface mass balance of the ice sheet is expected to become persistently negative (Noël et al. 2021); we thus remove floating ice as a crude representation of the hydrofracturing of the ice shelf as surface ponding drastically increases, as has been observed in Antarctica (e.g., Scambos et al., 2000; Banwell et al., 2013). While hydrofracture could become an important process earlier or later than 2055, this tipping point provides a reasonable choice

of timing for this sensitivity experiment that still allows us to match observations early in the simulation. This experiment was done for all climate forcings and both basal friction law exponents.

   In the second experiment, we examine the sensitivity of our century-scale projections to the assumed mean 2007–2017 sub-shelf melt rate by tuning sub-shelf melting to 10 and 30 m yr$^{-1}$ instead of 20 m yr$^{-1}$. This experiment was done for the CNRM and HadGEM2 climate forcings as the bounding cases of mass loss in our perturbed parameter ensemble (Figure 5),

with both basal friction law exponents.

   In the third sensitivity experiment, we explore the impact of our choice of 3 km yr$^{-1}$ maximum calving rate by including a set of simulations with 1, 5, and 7 km yr$^{-1}$ upper limits, as well as a simulation with no limit imposed on calving rates. These runs were done with the MIROC5 climate forcing, q = 1/7.

   In the fourth sensitivity experiment, we explore the impact of uncertainty in bed topography by repeating a set of runs

using BedMachine version 3 (Morlighem et al., 2017) for comparison with our main results using BedMachine version 4. This is motivated by the ~20–25% uncertainties in ice thickness reported near the northern portion of the marine margin in BedMachine version 4. While the differences between versions 3 and 4 are slightly larger than the uncertainty within version 4, we use this as a bounding case that still uses a data-constrained, mass-conserving bed that has been used in previous studies (Goelzer et al., 2020; Choi et al., 2021). We repeat both the optimization for the friction parameter $\mu$ and the basal

friction law exponent tuning procedure described above but applied to the bed topography from BedMachine version 3. We then run forward simulations from 2007–2100 for all three climate forcings and both basal friction law exponents.

### 2.6.4 Mesh convergence

Previous work with MALI has shown accurate simulation of grounding-line dynamics at 1 km resolution (Hoffman et al., 2018, 2019). However, these studies included buttressing ice shelves and did not use a pressure-dependent power-law basal

friction relationship. We performed a set of four simulations to determine the influence of mesh resolution on the modeled behavior of Humboldt Glacier using variable-resolution meshes with cell spacing of 0.25 – 10 km, 0.5 – 10 km, 1 – 10km, and 3 – 30km. We applied a Gaussian filter with a radius of 3 km to the BedMachine v4 bed topography and ice thickness datasets before interpolating to the meshes for these experiments, to ensure that model behavior is controlled by model physics and not simply by resolving features of the bed topography. All simulations were run from 2007–2050 using surface

mass balance and ocean thermal forcings from the MIROC5 RCP8.5 scenario. For the mesh convergence tests, submarine melt beneath ice shelves and at grounded margins are implemented as described above, calving is not applied, and we use a basal friction law exponent of q=1/5.





## 3 Results

### 3.1 Mesh convergence test

We find that our 1–10 km resolution mesh agrees well with results from the 0.25–10 km and 0.50–10 km meshes, while the 3–30 km mesh gives considerably different results in terms of changes in volume above floatation and grounded ice area (Figure S2). Thus, we choose the 1–10 km mesh for all simulations in the perturbed parameter ensemble as the best trade-off between accuracy and computing cost. We note that simulations using the von Mises stress calving law do not converge monotonically with increasing resolution, likely because of large stress gradients near the glacier terminus; the calving law

would likely have to be tuned independently for each resolution, making the results of a convergence test ambiguous. This convergence test does not examine the dependence of the solution on resolving the small-scale bed topography.

### 3.2 Perturbed parameter ensemble

### 3.2.1 Sea level contribution

The results of the perturbed parameter ensemble that constitutes our SLR estimates to 2100 are shown in Table 1 and Figures

4 and 5. All simulations predict significant retreat in the northern section of the terminus (Figure 4). Grounding-line retreat in our minimum mass loss simulations with a 2017 calving front position is nearly as extensive as those using von Mises stress calving, with variation due to climate forcing and basal friction law exponent. We find that for our fixed 2017 calving front scenarios, the model predicts 5.5–6.7 mm SLR by 2100, relative to our control simulations (~0.85 mm). In these minimum mass loss simulations, almost all the variability in SLR is due to the choice of climate forcing, with MIROC5 and

CNRM predicting ~5.5–6 mm SLR by 2100, and HadGEM2 predicting ~6.5 mm SLR by 2100. The two values of the basal friction law exponent (1/5 and 1/7) predict very similar SLR rates and totals over the whole century within each climate forcing. Thus, for our fixed 2017 calving front scenarios, the calibration of the basal friction law has successfully constrained the exponent to a range where the results are no longer sensitive to its uncertainty.

Using the von Mises stress calving parameterization leads to greater variability and higher mass loss than the

assumption of minimal calving after 2017. Our von Mises calving scenarios predict 6.5–9.2 mm SLR from 2007–2100 relative to our control simulations, with variability due to stress threshold tuning, basal friction law exponent, and climate forcing (Figure 5). We also include a set of six runs with a linear basal friction law (q=1) and low and high calving sensitivities to illustrate the importance of tuning the exponent to match observations. When compared with the linear basal friction law, our runs with tuned basal friction exponents predict up to twice as much SLR by 2100.



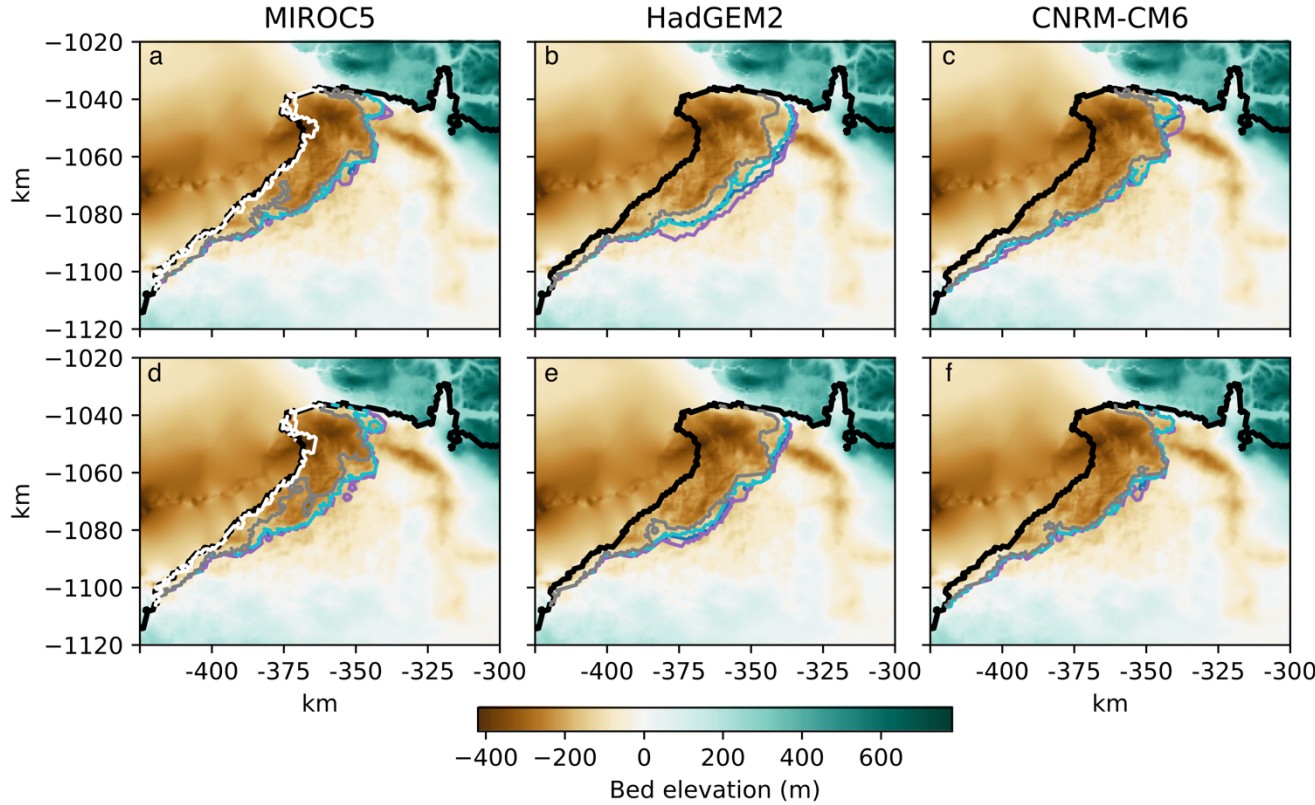

**Figure 4: Grounding line position at 2100 for all members of the perturbed parameter ensemble.** Results are shown for q=1/5 (a–c) and q=1/7 (d–f) basal friction law exponent. Black curves represent the initial ice extent in 2007. Grounding lines from control runs are shown in white; the same control run is used for all climate forcings with a given value of q. Curve colors represent calving scenario — grey: 2017 calving front; cyan: high $\sigma_{max}$ ; blue: medium $\sigma_{max}$; purple: low $\sigma_{max}$.

### 3.2.2 Ice shelf growth

We find that a significant amount of floating ice grows by the mid-century in all ensemble members using von Mises stress calving, and late in the century for most minimum mass loss calving scenarios (Figure 5; Figure S3; Figure S4). This is not generally anticipated for Greenland, where two out of only five total ice shelves have collapsed since the 1990s (Mottram et al., 2019) and model results predict the loss of the nearby Petermann Glacier ice shelf in the coming decades (Åkesson et al., 2021). Under the HadGEM2 climate forcing, the ice shelf is lost by the end of the century, but significant floating ice persists until 2100 for the majority of runs with MIROC5 and CNRM-CM6 climate. As expected, the volume of the ice shelf is dependent on calving tuning and basal friction law as well as climate, with greater volumes of floating ice predicted in runs with higher $\sigma_{max}$ and a more plastic bed rheology (q=1/7). However, in cases with HadGEM2 climate forcing, an increase in ice-shelf basal melting due to warmer ocean temperatures beginning in the early 2070s (Figure 6) removes the majority of floating ice within a decade. This period also corresponds with an increase in the rate of SLR contribution around





2075 for HadGEM2 climate (Figure 5) as the surface mass balance over grounded ice also becomes increasingly negative (Figure 6). We further explore the impact of this floating ice in two sets of sensitivity experiments, described below.

**Figure 5:** (a–c) Projections of volume loss and equivalent global mean sea-level rise from ensemble of 21st century simulations of Humboldt Glacier. Solid curves represent basal friction law exponent $q = 1/5$; dashed curves $q = 1/7$; dotted curves $q=1$ for comparison. The corresponding control run has been subtracted from each curve. (d–f) Projections of total floating ice volume from the same simulations as the top row, without the $q=1$ cases. Curve colors denote calving scenario. Values of $\sigma_{max}$ corresponding to low, medium, and high thresholds are found in Table 1.



### 3.2.3 Mass Budgets

We next investigate the contributions of individual processes (calving, basal and submarine melting, grounding-line flux, and surface mass balance) to the evolution of the total volume of both grounded and floating ice. Figure 6 shows the cumulative mass budgets for grounded (top row) and floating (bottom row) ice for the von Mises stress calving runs from our perturbed parameter ensemble through time. For grounded ice, we find that submarine melting at the grounded terminus (also called undercutting) contributes negligibly to the mass budgets over the 21st century, despite potentially being the dominant driver of recent retreat (Rignot et al., 2021). Undercutting does play a role in simulated mass loss over the historical period, but is not the dominant mode of mass loss; however, melt at the base of floating ice is also significant over this period. Because floating ice early in the model simulations is very close to the floatation thickness, our model could be over-estimating ice shelf melt and underestimating undercutting at the grounded front. We ran a set of six simulations from 2007–2021 with undercutting deactivated and found that the total calving flux is significantly decreased compared with simulations that include undercutting (Figure S5). This is because the grounded ice margin is close to its floatation thickness, so a small amount of undercutting causes the grounded margin to go afloat, which causes increased velocity and tensile stress, which in turn cause higher calving rates. Thus, while undercutting does not contribute significantly to the mass budget directly, its control on calving makes it an important process governing the retreat of Humboldt Glacier over the observational period.

Grounding-line flux is the largest contributor to total grounded ice loss in almost all simulations, but it is overcome by surface mass balance for the cases using HadGEM2 climate and q=1/5 basal friction law exponent. Calving from the grounded terminus is a significant contributor to grounded ice loss in all simulations, while melting at the base of grounded ice is a negligible term in the mss budget. The value of the basal friction law exponent makes the largest difference in grounding-line flux and a minor difference in calving from the grounded terminus; it makes almost no difference in surface mass balance, basal melting, or undercutting contributions to the budget. Net surface mass balance contribution is almost entirely negative for MIROC5 and HadGEM2, while the CNRM-CM6 net surface mass balance is positive until the mid-2060s, after which it becomes steeply negative. Surface mass balance contributes ~3.1 mm SLR for CNRM-CM6, ~3.7 mm SLR for MIROC5, and ~4.1 mm SLR for HadGEM2 climate forcing. Other terms in the grounded budget in Figure 6 should not be interpreted directly in terms of SLR because much of the ice lost by calving, grounding line flux, and undercutting is not above flotation.

Calving and sub-shelf melting dominate mass loss from floating ice. Calving is dominant early in the century, while sub-shelf melting generally reaches a similar magnitude by the end of the century. Surface mass balance plays a minor role on floating ice. We observe higher sub-shelf melting, calving, and grounding-line flux in the q=1/7 runs than in the q=1/5 runs, but these roughly balance out to give similar net volume change for floating ice between the two basal friction law exponents.

**Figure 6: Mass budgets for grounded (a–c) and floating (d–f) ice for all runs using von Mises stress calving.** Colors indicate components of the budget, while line styles indicate the value of the basal friction law exponent (solid: q=1/5; dashed: q=1/7). Insets in each panel show the 2007–2021 period. GL flux: grounding-line flux.

## 3.3 Sensitivity Experiments

### 3.3.1 Removal of floating ice

We find that the presence of floating ice significantly reduces mass loss from Humboldt Glacier for all three climate forcings and both basal friction law exponents (Figure 7). Simulations with floating ice removed in 2055 and prevented from regrowth predict 1–2 mm (up to ~25%) more SLR by 2100 than the counterpart simulations within the main perturbed





parameter ensemble, due to a combination of loss of buttressing and increased calving from the grounded ice margin. Removing floating ice at the beginning of the simulation would likely lead to greater SLR contribution, but it would not be possible to match the observed retreat and speed-up.

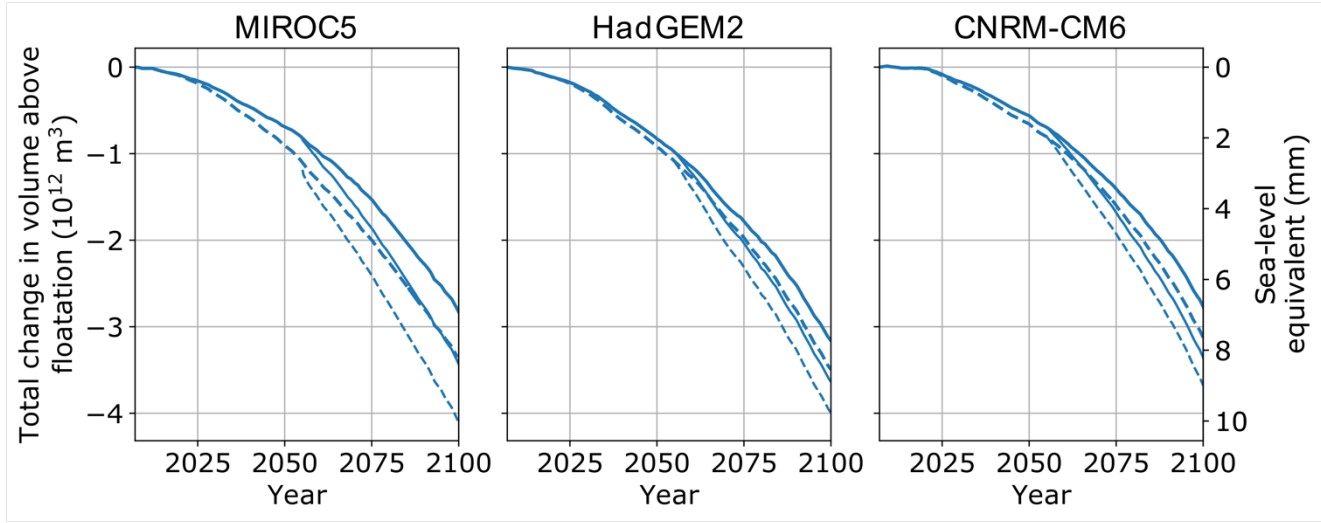

**Figure 7: Results of sensitivity test in which floating ice is removed in 2055 and not allowed to regrow.** As in other figures, the solid and dashed curves represent q=1/5 and q=1/7, respectively. Thicker upper curves are the runs with floating ice maintained (i.e., same as blue curves in Figure 5); thinner lower curves are with floating ice removed.

### 3.3.2 Sub-shelf melt tuning

Figure 8 shows the results of our sensitivity experiment with mean 2007–2017 melt tuned to 10, 20, and 30 m yr$^{-1}$ for the CNRM and HadGEM2 climate forcings. We find that our assumptions about melt rates over the historical period have a small effect on our retreat projections. For the HadGEM2 climate forcing, all simulations with the same basal friction law exponent predict the same SLR by 2100 to within ~0.7 mm; for CNRM they agree to within ~0.5 mm. For both climate forcings, there is a weakly non-linear effect of the sub-shelf melt tuning, in which the 10 m yr$^{-1}$ tunings lead to slightly more

mass loss than the 20 m yr$^{-1}$ tunings (Figure 8 insets); this occurs for all of the climate–basal friction pairs except for HadGEM2, q=1/7. However, the differences are likely small enough to be negligible compared with the uncertainties inherent in the melt parameterization.



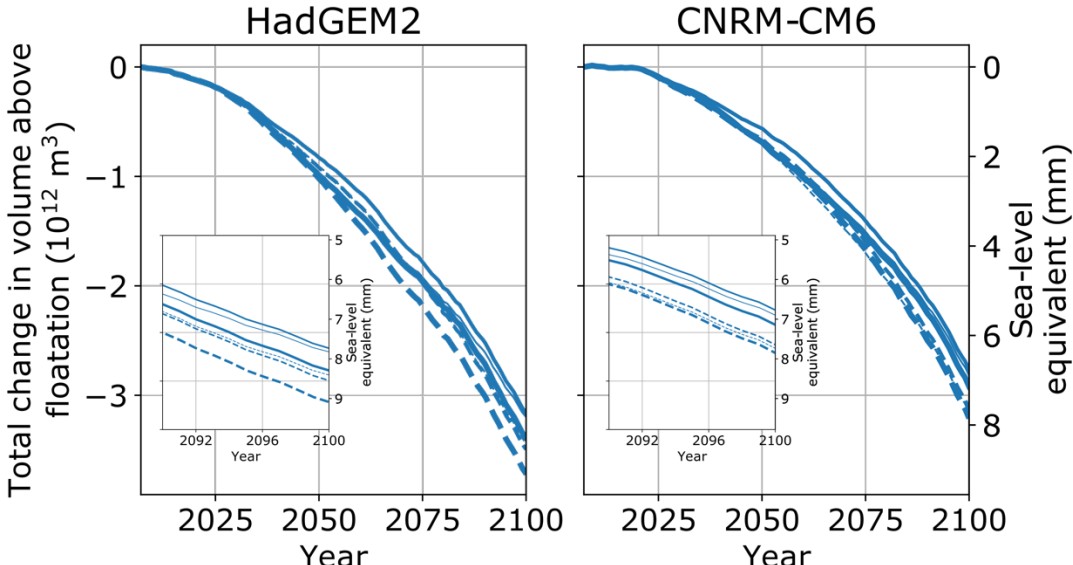

**Figure 8: Results of ice shelf melt sensitivity test.** As in other figures, the solid and dashed curves represent q=⅕ and q=1/7, respectively. Line thickness corresponds to sub-shelf melt rate tuning. Thin: 10 m yr[-1]; medium: 20 m yr[-1]; thick: 30 m yr[-1]. The 20 m yr[-1] melt runs are the same as the corresponding curves in Figure 5. Insets show the last ten years of the simulation.

### 3.3.3 Calving rate limit

Figure 9 shows the results of our sensitivity experiment with five choices of upper limits on iceberg calving rate: 1, 3, 5, and 7 km yr[-1], as well as one with no upper limit. We find that removing this upper limit leads to unreasonably high sustained glacier velocities of 35–40 km yr[-1], and brief surges in this range occur even with the 7 km yr[-1] calving rate limit . Thus, we consider ≤5 km yr[-1] to be the reasonable range of upper limits on calving rates for Humboldt Glacier using the von Mises stress calving parameterization and our semi-plastic bed rheology. While our choice of a 3 km yr[-1] calving rate limit results in a ~15% uncertainty in sea-level contribution in this case, this choice could combine non-linearly with choices of climate forcing, calving scenario, and basal friction law, leading to a larger overall uncertainty. We also note that the extent of the ice shelf is greatly reduced for assumed maximum calving rates of ≥5 km yr[-1]. Because a maximum calving rate of 5 km yr[-1] still gives reasonable results, it is possible that the significant ice shelf growth is a consequence of the assumed 3 km yr[-1] maximum calving rate.

We have used the calving rate limit of 3 km yr[-1] for the perturbed parameter ensemble because this limit was used in previous work (Choi et al., 2018) and because we are attempting to mitigate the positive feedback between calving rate and ice flow speed. It is not clear at what point this feedback becomes unphysical, even if modeled ice velocities remain within the range observed across the GrIS. However, we realize that our assumption may lead to a conservative estimate of SLR. To address this, we have run an additional set of three simulations using all three climate forcings, q = 1/7, low $\sigma_{max}$, and a


calving rate limit of 5 km yr⁻¹ to estimate an upper bound on mass loss from Humboldt Glacier by 2100 (Figure S6). We find

an upper bound of ~11–12 mm SLR, depending on climate forcing, compared with ~8–9 mm using the 3 km yr⁻¹ limit on calving.

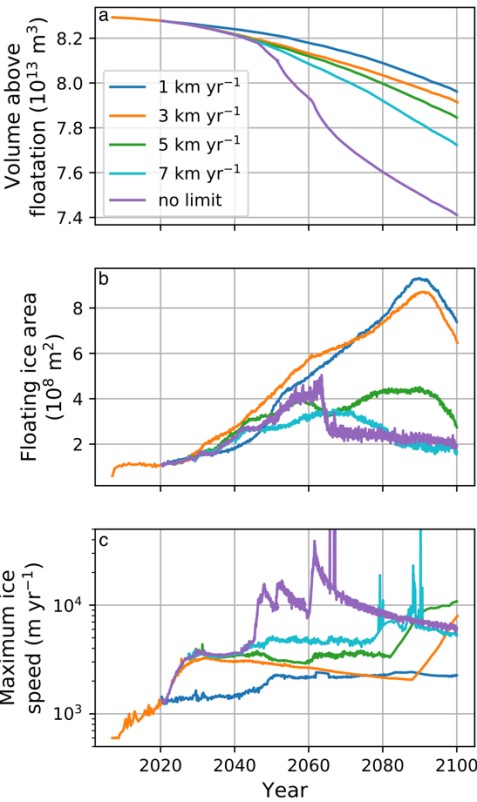

**Figure 9: Results of calving rate limit sensitivity test.** Line colors correspond to imposed calving rate limits. Based on the maximum modeled glacier speed in panel c, we take ≤5 km yr⁻¹ to be a reasonable range.

**3.3.4 Bed topography**

The new bathymetry data integrated into BedMachine version 4 result in significantly thicker ice at the glacier margin than in BedMachine version 3 (Rignot et al., 2021; Morlighem et al., 2017, 2021). We find that for the version 3 bed, the model requires a considerably more plastic basal friction law to match the observed 2007–2017 speedup. We find exponents in the range of 1/10–1/25, compared with 1/5–1/7 with the version 4 bed. Because of this more-plastic bed rheology, our century-

scale simulations with the version 3 bed predict a higher sea-level contribution by 2100 (Figure 10). In general, the BMv3 simulations lose less mass early in the century, but all the BMv3 simulations surpass their BMv4 counterparts in terms of SLR contribution in the latter part of the century, due to increased ice flux allowed by the more-plastic basal friction law. We also find that simulations using the BMv3 bed topography exhibit ice-shelf growth later in the century than in the corresponding BMv4 bed topography simulations.





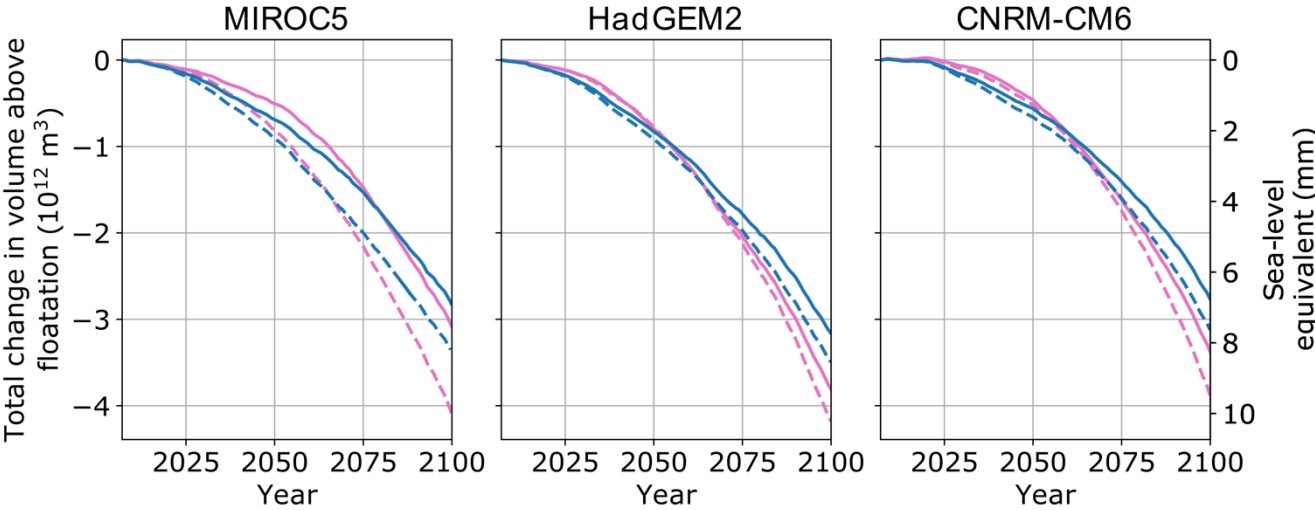

**Figure 10: Results from bed topography sensitivity test.** Line colors correspond to the bed topography product version. Blue: BedMachine v4; pink: BedMachine v3. As in other figures, the solid and dashed blue curves represent q=⅕ and q=1/7, respectively; for pink, solid and dashed are q=1/10 and q=1/25, respectively.

## 4. Discussion

### 4.1 Controls on historical retreat of Humboldt Glacier

We find that melt undercutting at the grounded glacier terminus is a significant contributor to the predicted retreat over the observational period through its control on iceberg calving (Figure 6; Figure S5). However, the amount of submarine melt does not directly dominate the retreat as found by Rignot et al. (2021). This discrepancy could be due to the fact that we use model-based thermal forcing fields provided by ISMIP6 to calculate melt, rather than directly using CTD data. The existence of some floating ice in our simulations over the historical period also leads to basal melt rather than melting of the grounded terminus in these areas. We also do not take into account the "calving multiplier effect" by which melt undercutting drives higher calving rates even when it does not directly cause grounded ice to go afloat (O'Leary and Christofferson, 2013; Benn et al., 2017), which would depend on resolving the geometry of the face of the glacier terminus (Slater et al., 2021). This lack of an amplified influence of submarine melting on calving could be one reason for the small values of $\sigma_{max}$ necessary to initiate retreat. In other words, $\sigma_{max}$ may need to be small in the model to make up for the unresolved process of higher tensile stresses induced by undercutting.

Our model does not reproduce the seasonal velocity cycle observed at Greenland outlet glaciers, which are often strongly controlled by surface runoff to the subglacial hydrologic system (Hoffman & Price, 2014; Moon et al., 2014) or by changes in buttressing due to melange and ice tongue formation and break-up (Joughin et al., 2012). The seasonal velocity cycle observed at Humboldt Glacier is strong, with velocities temporarily doubling or tripling near the terminus in summer



compared with winter (Joughin et al., 2010; 2021). This seasonal cycle is superimposed on the longer-term tripling of velocity near the terminus and a 50% increase in velocity ~25 km up-glacier of the terminus between 2012 and 2017 (Figure 1; Joughin et al., 2018), during which time the terminus retreated 1–3 km in the north, and <1km in the south (Joughin et al., 2015, 2017; Moon & Joughin 2008). We attempt to reproduce this longer-term acceleration by tuning the basal friction law

exponent and the von Mises threshold stress parameter. Reproducing the full seasonal cycle would likely require coupling our ice dynamics model to a subglacial hydrology model, which is currently a major challenge for ice sheet simulations (Flowers et al., 2018). The summer acceleration is accompanied by terminus retreat at other Greenland outlet glaciers, as higher extensional strain rates promote calving (Kehrl et al., 2017). Thus, we may need to use the small value of $\sigma_{max} \leq 200$ kPa because we do not reproduce the high summer velocities, strain rates, and tensile stresses that induce most calving in

reality.

In a sensitivity analysis of a flowline model, Carr et al. (2015) found that the greater retreat of the northern section compared with the southern section of the Humboldt Glacier terminus could be partially due to sea-ice buttressing and meltwater depth in crevasses. They found that rising air temperatures (and thus presumably increased meltwater flux into crevasses) and a decrease in sea-ice concentration coincided with the initiation of its rapid retreat after 1999. We have not

represented these processes in our model, yet we are able to reproduce the historical behavior relatively well. Because of the numerous differences between our model and theirs (bed topography, number of spatial dimensions, stress balance approximation, plasticity of basal friction law, choice of calving parameterization, etc.), a direct comparison to their results is not feasible. However, we note that Carr et al. (2015) use a basal friction law exponent of q=1/3, while we find $1/7 \leq q \leq 1/5$ necessary to match observed changes. They also do not include submarine melting in their model experiments, concluding

that it is likely a small contribution. In broad agreement with the results of Rignot et al. (2021), we have found that submarine melt is in fact an important factor in the retreat of Humboldt Glacier (Figure 6, Figure S5). However, if meltwater in crevasses and sea-ice concentration are indeed primary controls on the past and present retreat of Humboldt Glacier, then it could become more sensitive to surface and ocean temperatures as the climate warms than we have accounted for here. Whether this increased sensitivity would lead to significantly more sea-level rise from Humboldt than we estimate here is an

open question.

**4.2 Comparison to previous 21st century projections from Greenland**

Our estimated SLR contribution of 5.5–9.2 mm by 2100 is considerably higher than a recent estimate of ~3.5 mm from Humboldt Glacier (Choi et al., 2021). We attribute this primarily to our calibration of the basal friction law to match observed surface velocity changes. Choi et al. (2021) used a linear-viscous treatment of basal friction and did not attempt to

reproduce the speed-up over the historical period, likely because they modeled the entire GrIS. In their projection for Humboldt Glacier, the contribution from surface mass balance was about 50% larger than the dynamic contribution by 2100. In our von Mises stress calving simulations, ice dynamics (including both ice flow and calving) is the larger direct contributor in most cases (Figure 5, 6), although some of the ice dynamical response is also driven by surface mass balance.





While our estimate differs from theirs, our findings support their conclusion that ice dynamics will continue to be an
important part of the GrIS mass budget through the 21st century.

Our projected SLR contribution is also a sizable fraction of the $90 \pm 50$ mm SLR from the whole GrIS by 2100 predicted
by the ISMIP6 multi-model ensemble for the RCP8.5 emissions scenario (Goelzer et al., 2020). There are too many degrees
of freedom between the many models and modeling choices used in the ISMIP6 whole-GrIS ensemble to determine whether
our basal friction law tuning is the reason for this, but we suspect that it is a primary contributor because our tuned basal
friction law exponent leads to up to twice as much mass loss when compared with an assumed linear viscous basal friction
law (Figure 5). However, because Humboldt Glacier represents ~5% of the area of the Greenland Ice sheet, our lower bound
of 6.5 mm from the von Mises calving simulations represents a proportional contribution when compared with the upper
value of 140 mm from ISMIP6 for the whole GrIS (Goelzer et al., 2020). If the behavior we find here extends to other major
outlet glaciers (e.g., Åkesson et al., 2021), then the ISMIP6 projections from models using linear or Weertman basal friction
relationships may systematically underestimate 21st century retreat and mass loss.

It is unclear whether the higher climate sensitivity of the CMIP6 models compared with CMIP5 represents an
improvement in climate forecasting (Edwards et al., 2021; Payne et al., 2021). The single CMIP6 climate forcing we include
here (CNRM-CM6) predicts high positive surface mass balance towards the glacier interior, resulting in a net positive
surface mass balance contribution for the first half of our simulations, while the two CMIP5 climate forcings predict net
negative surface mass balance contribution through the entire century (Figure 6). However, the steeper negative slope of the
CNRM-CM6 forcing by the end of the century indicates that the 22nd century could see extreme negative surface mass
balance dominating mass loss (e.g., Goelzer et al., 2013). Other CMIP5 and CMIP6 Earth system models and high resolution
regional climate modeling (not examined here) agree on a net negative surface mass balance for the GrIS after about 2055
(Noël et al., 2021). Thus, while we find a slightly larger direct SLR contribution from ice dynamics than from surface mass
balance, these relative magnitudes could change after 2100. However, because surface mass balance influences ice dynamics
through its various effects on basal friction (changing both ice overburden and basal water pressure in reality), these two
contributions cannot be completely disentangled.

### 4.3 Influence of the basal friction law

Our choice of an effective pressure-dependent power-law (Budd-type) basal friction relationship was motivated by the need
to match the observed tripling of flow speed at the northern section of the marine terminus in the observational period
(Figure 1, 3), for which a linear-viscous basal friction law is not suitable. However, the proper representation of the effective
pressure term in this relationship is highly uncertain. As in many previous studies, we use the assumption that the subglacial
hydrologic system is perfectly connected to the ocean (e.g., Vielli et al., 2001; Seroussi et al., 2013; Asay-Davis et al.,
2016; Morlighem et al., 2019). However, this assumption can lead to effective pressures that increase too rapidly with
distance from the grounding line when compared with subglacial hydrology model output (Smith-Johnsen et al., 2020; Hager
et al., 2021). While this problem is offset by the basal friction parameter μ in the optimization, long-term projections are





complicated by the resulting spatial pattern, which likely leads to values of μ that are inappropriately low when the grounding line has retreated far inland. We do not expect this to be a significant issue in our century-scale runs, as the grounding line only retreats ~30 km in our most aggressive scenario (q = 1/7, HadGEM2 climate, low $\sigma_{max}$) (Figure 4),

where μ is still relatively high and effective pressure in the initial condition is relatively low. In the absence of a subglacial hydrology model coupled to the ice dynamics model, better treatments of effective pressure are not readily available. However, simple parameterizations of effective pressure that produce the lower gradients observed in subglacial hydrology model results could improve the reliability of long-term projections.

While we have explored the parameter uncertainty within our chosen Budd-type basal friction relationship (Budd et al.,
1979; 1984) by using a range of power-law exponents, we have not explored uncertainty in the basal friction coefficient, $\mu$, from our optimization. Because $\mu$ is a spatially variable field, quantifying this uncertainty would require advanced and expensive approaches, such as Bayesian inference with Markov Chain Monte Carlo sampling (see, e.g., Petra, 2014), which are beyond the scope of this study. The uncertainty in $\mu$ depends on uncertainties in the surface velocity data, the stress balance approximation and enthalpy model, uncertainty in the bed topography, and other model parameters.

We have also not explored the structural uncertainty inherent in the choice of the form of the basal friction law. Other commonly-used basal friction relationships include (1) a linear viscous relationship with or without effective pressure dependence; (2) a power-law relationship (with q typically taken to be 1/3) that is not dependent on effective pressure (Weertman, 1957); (3) a fully plastic till strength Coulomb law used for soft beds (Tulaczyk et al., 2000a,b); (4) a regularized Coulomb relationship in which the influence of effective pressure reduces with distance from the grounding line
(Schoof, 2005; Joughin et al., 2019); and (5) a combined treatment with a sharp transition between Coulomb and power-law friction (Tsai et al., 2015). Of this list, our chosen semi-plastic Budd-type law and laws 3–5 above are appropriate for representing fast flow in ice sheets. The linear viscous relationship is technically valid only for regelation-dominated basal flow, which is in general much slower than enhanced creep (Weertman, 1957; Fowler, 2010). The Weertman (1957) law is in general considered valid for hard beds with basal flow dominated by enhanced creep, while sliding beneath ice sheets is
generally considered to be strongly dependent on till deformation and subglacial water pressure (e.g., Tulaczyk et al., 2000a,b; Zoet & Iverson, 2020). However, both the linear viscous and Weertman laws are still widely used for ice-sheet scale simulations, despite being largely inappropriate for representing fast ice flow. For example, six of the fifteen models participating in the Antarctic Buttressing Model Intercomparison Project use a Weertman-style basal friction relationship, and another three use a linear viscous relationship (Sun et al., 2020). The basal friction relationships used by the models
participating in ISMIP6 Greenland are not systematically reported, but six of the eight reported (out of thirteen total) use a linear relationship or a power-law relationship with an exponent of 1/3 (Goelzer et al., 2020). More widespread use of observed temporal speed changes to calibrate the basal friction relationship would likely lead modelers to abandon the Weertman and linear-viscous laws in large scale ice sheet model simulations (e.g., Gillet-Chaulet et al., 2016; Joughin et al., 2019).





Our tuning of the basal friction law exponent underscores the point that optimizing the basal friction parameter can give a good fit to observations at the initial condition for most choices of the basal friction law, but that thereafter the evolution diverges widely based on this choice (Gillet-Chaulet et al., 2016; Nias et al., 2018; Brondex et al., 2019; Joughin et al., 2019; Åkesson et al., 2021). Hindcasting simulations used to tune the basal friction law to reproduce observed changes are thus necessary to reduce uncertainty in projections of future behavior. However, there does not seem to be any consensus on the

best form of the basal friction relationship, and the best choice could be glacier- or basin-dependent. Additionally, the regularized Coulomb and fully plastic till strength relationships have additional unknown parameters that either need to be optimized by inverse methods, tuned in forward runs, or explored in sensitivity tests, and which in reality would likely exhibit wide spatial variation.

The form of the basal friction relationship often exerts a strong control on modeled glacier behavior (Brondex et al.,

2017, 2019; Nias et al., 2018; Joughin et al., 2019; Åkesson et al., 2021), but there seems to be no rule of thumb for determining *a priori* the relative magnitudes of mass loss between the Budd-type and regularized Coulomb basal friction laws. For instance, Brondex et al. (2019) found that Pine Island and Thwaites glaciers thinned and retreated less with the Budd law than with the Coulomb law, while Cosgrove and Dotson glaciers retreated considerably more with the Budd law than with the Coulomb law. Åkesson et al. (2021) also found that the Budd law led to greater mass loss than the regularized

Coulomb law at Petermann Glacier, even though these two gave the best fits to observations of the laws they examined. Thus, it appears that the relative influence of these basal friction relationships on mass loss may be dependent on bed or ice geometry, and so we cannot confidently state whether using regularized Coulomb friction law would lead to more or less mass loss than we predict using the Budd-type law.

### 4.4 Influence of the iceberg calving law

We find that the von Mises stress calving law leads to a positive feedback between calving and ice velocity, which is exacerbated by our non-linear basal friction law (Figure 9). The imposed upper bound of 3 km yr$^{-1}$ on the calving rate is an ad hoc measure to prevent the runaway feedback between basal friction, ice velocity, and calving rate. While previous studies have used an upper bound on iceberg calving for this and other calving laws (e.g., DeConto & Pollard, 2016; Choi et al., 2018), this is a problem that should be rectified with improved calving parameterizations and calving fronts resolved in

three dimensions, both of which pose significant challenges. The issues we find with the von Mises stress calving law lend some support to the suggestion that calving front position parameterizations should be used instead of calving rate parameterizations (Amaral et al., 2020). However, many existing calving front position parameterizations are without a convincing physical basis (for example, the calving laws based on height or thickness above floatation), while calving rate parameterizations like the von Mises stress formulation are desirable because they take the stress and strain rate fields into

account. The crevasse depth model (Benn et al., 2007; Nick et al., 2010) generally recommended by Amaral et al. (2020) could be a good alternative to the von Mises calving law. However, Amaral et al. (2020) found that it performed worse than the von Mises stress calving law at Humboldt Glacier. It is also unclear how the crevasse water depth tuning parameter





should evolve in time as surface melt increases. We have not used it here for these reasons, but we acknowledge that it could alleviate some of the problems we find with the von Mises calving law, which could potentially justify accepting a decreased
ability to match observed retreat rates.

We use a single spatially uniform value of $\sigma_{max}$ in each simulation, as in previous work (Morlighem et al., 2016, 2019; Choi et al., 2018, 2021). Another option would be to use different values of $\sigma_{max}$ for grounded and floating ice (Choi et al., 2017), but because calving from the grounded margin is a relatively small contribution to the mass budget (Figure 6), this is unlikely to have much impact on our results while greatly increasing the complexity of our calibration step. In reality $\sigma_{max}$
would likely be dependent on numerous factors such as ice fabric, temperature, damage, and meltwater availability for hydrofracture, all of which will vary in space and time. If $\sigma_{max}$ was allowed to evolve as some function of these properties, there could be potential for a negative feedback on calving as the calving front retreats into regions of stronger, less damaged ice or areas with less meltwater available. Whether this effect would be large enough to significantly impact our results is an open question.

Our simulations using the von Mises stress calving law require $\sigma_{max} \leq 200$ kPa in order for Humboldt Glacier to retreat, while Amaral et al. (2020) found a $\sigma_{max}$ value of ~2 MPa. We can identify a number of possible explanations for this discrepancy. First, Amaral et al. (2020) calibrated $\sigma_{max}$ along a 2 km wide flowband from 2012–2014, while we calculate the misfit across the full glacier terminus from 2007–2017, and we attempt to match both observed velocities and terminus positions. Second, our model does not reproduce the observed seasonal velocity cycle (Moon et al., 2014), while Amaral et
al. (2020) used observations from May 2012 and June 2014, months on either side of the beginning of the summer speedup. For example, the velocity within 10 km of the terminus along the flowband analyzed by Amaral et al. (2020) doubled in the month between May and June 2014 (Joughin et al., 2020). This kind of seasonal acceleration is accompanied by terminus retreat at Helheim and Kangerlussuaq glaciers (Kehrl et al., 2017), and Humboldt could behave in a similar way. This could explain why observations that span the shoulder of the seasonal cycle (May and June) would require a higher $\sigma_{max}$ than is
necessary for us to match mean observed retreat rates in our model: higher strain rates during seasonal acceleration result in a higher calculated von Mises stress, which in turn requires a higher value of $\sigma_{max}$ to match retreat. Finally, as noted above, we do not account for the calving multiplier effect, which could greatly increase tensile stresses behind undercut portions of the terminus. Given the considerable differences between our modeling and Amaral et al.'s (2020) observationally constrained approach — including different ice temperature fields, bed topography, resolutions, and time periods — we find
this a sufficient reconciliation of the discrepancy in $\sigma_{max}$. However, this illustrates that care should be taken when using values of tuning parameters from observational approaches directly in modeling studies, as the model may be missing physics or forcings that are implicitly accounted for in observations. As most ice sheet models do not reproduce the seasonal velocity cycle observed at Greenland outlet glaciers, the timing of observations used in model calibration and validation is of critical importance. Thus, we posit that $\sigma_{max}$ should be seen as a tuning parameter without much physical significance, and
thus it may not fall within the reported range of ice tensile strength (e.g., Petrovic, 2003).



## 5. Conclusions

We have presented an ensemble of model simulations beginning from an optimized initial condition of Humboldt Glacier, North Greenland, through the 21st century that are consistent with observed retreat rates and velocity changes from 2007–2017. We used three different climate forcings, spanning RCP8.5 and SSP5-8.5 scenarios from CMIP5 and CMIP6 models.
Our main perturbed parameter ensemble explores the uncertainty in iceberg calving, climate forcing, and non-linearity of the basal sliding law, within the constraints provided by recent observations. In addition, we tested the sensitivity of the model to ice-shelf growth, sub-shelf melt rates, maximum calving rates, and bed topography.

The results of our perturbed parameter ensemble indicate that Humboldt Glacier is likely to contribute 6.5–9.2 mm to global mean sea level by 2100 under warm climate scenarios (RCP 8.5 and SSP5-8.5), with variability due to the choice of climate model, calving law tuning, and basal sliding law. Our minimum mass loss estimates — which assume no additional retreat from calving after 2017 — range from 5.5–6.7 mm depending mainly on the choice of climate forcing. We find a relatively small range of acceptable tuning parameter values for the von Mises stress calving law (160–180 kPa or 170–190 kPa, depending on the climate forcing and basal friction law), but this leads to up to ~1 mm variation in predicted sea-level rise contribution by 2100. These estimates are significantly higher than results from our own model using a linear basal friction law (4–5 mm), as well as another recent estimate using a different ice sheet model also with a linear basal friction law (~3.5 mm; Choi et al., 2021). We note that a positive feedback between ice velocity and calving rates in the von Mises stress calving law results in considerable uncertainty in SLR estimates because an assumed maximum calving rate must be imposed. For reasonable assumptions of maximum calving rates (≤5 km yr$^{-1}$ based on our sensitivity experiments), this could be as large as 30%. Improved calving parameterizations are needed to reduce uncertainty in SLR estimates over timescales relevant to present-day policy decisions.

Our ensemble predicts the growth of a significant ice shelf by the middle of the 21st century in most simulations, which is still present by the end of the century for about two-thirds of cases. While this is not generally anticipated for Greenland outlet glaciers, it is a robust feature of the model given that it occurs across many parameter combinations, although it is sensitive to the assumed maximum calving rate. In sensitivity experiments with the ice shelf removed in 2055, Humboldt Glacier contributed up to 2 mm more SLR by 2100, a maximum increase of ~25%. We also find that our projected SLR estimates are relatively insensitive to our choice of sub-shelf melt tuning.

Finally, we find that mass loss projections are strongly dependent on the bed topography data product used in simulations. Using BedMachine v3 bed topography instead of v4 yields up to an additional 2 mm SLR by 2100 for the cases we examined, a maximum increase of 25%. This is because the shallower bed in BedMachine v3 requires a much more plastic basal friction law (exponents of 1/10–1/25 for v3, compared to 1/5–1/7 for v4) to match observed velocity changes. Thus, uncertainty in the ice thickness near the Humboldt Glacier terminus — and potentially at many other GrIS outlet glaciers — could be a large source of uncertainty in estimates of future SLR. This indicates that forthcoming attempts to



estimate the future SLR contribution from the GrIS (e.g., ISMIP7) should explore the propagation of the uncertainty in bed topography and ice thickness.

Our findings suggest a number of other ways to improve confidence in SLR projections from ice sheet models. Systematic calibration of basal friction and iceberg calving parameterizations to historical observations of both velocity and ice extent changes are necessary. For models of the whole GrIS, this may have to be done by individual basin or sector, rather than using a single parameterization for the whole ice sheet. Calibrating the basal friction relationship to observed speed changes will likely lead to the abandonment of the commonly used linear viscous and Weertman basal friction

relationships in favor of more physically based parameterizations, which could lead to substantially higher estimates of SLR by 2100. Thus, current estimates of SLR from the GrIS this century could be substantially too conservative due to the numerous and significant challenges involved in calibrating models of the whole ice sheet. Parameterizations of iceberg calving continue to improve, but they are still a primary source of uncertainty. Finally, improved parameterizations of effective pressure are likely necessary for longer-term projections involving significant grounding-line retreat.

In summary, we find that Humboldt Glacier is likely to make a significant contribution to SLR this century and that, to-date, its potential contribution has been underestimated. While the precision of our estimate is limited by the current state of understanding of iceberg calving, basal sliding, bed topography, and the magnitudes of future warming of the ocean and atmosphere, all of our model simulations predict continued retreat of Humboldt Glacier that will lead to the deglaciation of the majority of its deep marine-based area by 2100. Our findings indicate that ice dynamics, rather than surface mass

balance, could be the dominant term in the mass budget of Humboldt Glacier this century.

*Code and data availability.* Model input, output, code, and analysis scripts are publicly archived and available online at https://doi.org/10.5281/zenodo.6338400 (Hillebrand et al., 2022). The archive contains annual 2D and 3D model output that can be visualized using ParaView (https://www.paraview.org). Scalar outputs are available for every timestep. 2D and 3D

model output for each timestep can be made available upon request, but are not necessary to reproduce the analysis or figures. The versions of the MALI and Albany code used in this study can also be found on GitHub:
https://github.com/MALI-Dev/E3SM/tree/d0c286778a6a035c600af891a268ed9f843aac52
https://github.com/sandialabs/Albany/tree/ecbc0182015fda8d6a2bb29eaa3a47c260f40b13

*Author contributions.* IH, SFP, TRH, and MJH conceived of the study. TRH and MJH designed the model experiments. TRH implemented calving and melting parameterizations in the model code, performed the model simulations and analysis, and wrote the manuscript with input from all authors. MJH, TRH, SFP, and MP developed the ice sheet model. MP performed the optimization for the initial condition. IH provided expertise in the use of remote sensing data.

*Competing interests*. The authors declare they have no conflict of interest.



*Acknowledgements.* This work was supported by the Probabilistic Sea Level Projections from Ice Sheet and Earth System Models project (https://doe-prospect.github.io/) and the Energy Exascale Earth System Model (E3SM) project (https://e3sm.org/) funded by the Biological and Environmental Research (BER), Advanced Scientific Computing Research

(ASCR), and Scientific Discovery through Advanced Computing (SciDAC) programs within the U.S. Department of Energy's Office of Science. Simulations were performed on machines at the National Energy Research Scientific Computing Center, a U.S. Department of Energy Office of Science User Facility located at Lawrence Berkeley National Laboratory, operated under Contract No. DE-AC02-05CH11231 using NERSC award using NERSC award ERCAP0016816. We also used computing resources at the Los Alamos National Laboratory Institutional Computing Program, which is supported by

the U.S. Department of Energy National Nuclear Security Administration under Contract DE-AC52-06NA25396. I.H. was supported by grants 80NSSC18K1027 and NNX13AI21A from the National Aeronautics and Space Administration. We thank Michael Kelleher for assistance with data processing.

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
