# Peer review of "The contribution of Humboldt Glacier, North Greenland, to sea-level rise through 2100 constrained by recent observations of speedup and retreat"

_The Cryosphere, 2022_

## Author Comment (AC1)

Response to reviewers' comments on "The contribution of Humboldt Glacier, North Greenland, to sea-level rise through 2100 constrained by recent observations of speedup and retreat" by Trevor R. Hillebrand, Matthew J. Hoffman, Mauro Perego, Stephen F. Price, and Ian M. Howat

We thank both reviewers for their constructive and thorough comments. Below, we reply point-by-point to the reviewer comments. Reviewer comments are in black, and author responses are in blue.

Reviewer 1:

Authors present numerical simulations of mass loss from Humboldt Glacier. Historical runs and observations (2007-2017) are used to constrain parameters in the description of basal rheology and the calving law. Optimal model parameters are used to produce projections of mass loss for a range of future forcing scenarios (2017-2100). Results highlight the importance of the basal sliding exponent, and best estimates of mass loss exceed previous projections by about a factor of 2.

The objectives and methodology of this study are clear; the results novel and well-presented; the conclusions well-supported, and overall this work is a good fit to The Cryosphere. I recommend publications with scope for some minor revisions, as outlined below.

The experimental design is mostly straightforward and easy to follow, though I wonder about the need to distinguish between the perturbed parameter ensemble, and the additional sensitivity experiments. After all, these are all sensitivity experiments, and personally I think the distinction overcomplicates the structure of the paper. An overview of the sensitivity to all physical parameters in a single Table would be nice. Some experiments that do not test the significance of physical parameters, such as the mesh resolution and potentially, bed topo, could be included in an Appendix to reduce the amount of information in the main text.

Author response: We agree that the distinction between the perturbed parameter ensemble and the sensitivity experiments needs to be made more clear. The perturbed parameter ensemble explores the parameter space that we are able to calibrate against observations, namely calving and basal friction law. The sensitivity experiments encompass processes and characteristics for which there is deeper uncertainty and that we cannot calibrate within our framework, such as submarine melt, ice-shelf collapse, maximum calving rates (which are likely much greater than anything in the observational period), and bed topography. We will make this distinction clearer when the experiments are introduced. We will move the mesh convergence test to the supplemental information as you suggest. The bed topography sensitivity experiment we choose to keep in the main text because the sensitivity to uncertainty in bed topography is a major conclusion of the paper that is likely applicable to many other major outlets.

The validation approach is interesting, though I miss a more in-depth description/motivation of the validation criteria. Fig3b suggests that the optimal choice of q is critically dependent on the velocity itself, so why choose 1/5-1/7, which only provides a best match for u>600m/yr? In this

regard, you might find the discussion in section 3.3 of [De Rydt et al. 2021] of interest, where authors show that the optimal sliding exponent for Pine Island Glacier is spatially heterogeneous. On a related note, I wonder if you can show the difference between observations and model in Fig3a, rather than the absolute model speed.

Author response: We chose the range of 1/5 to 1/7 not because it provided the best match to fast flow (q=⅛ provides the best match in the fastest velocity band), but because it provides a reasonable match across all three velocity bands. It is true that the optimal value of the sliding exponent is very likely spatially heterogeneous due to spatially varying bed characteristics, but the majority of ice sheet models are not currently capable of optimizing this value as a spatially varying parameter, which would require a transient optimization to invert for both mu and q. Thus, our range of 1/5–1/7 likely represents a compromise between the need for a larger exponent for slow-flowing regions and a smaller exponent for fast-flowing regions. We will update the text to more clearly explain the validation criteria. We will also include a panel showing the difference between model and observations, either in Fig 3 or in the supplement.

I think some further details about the melting and calving paramterization would be instructive for readers less familiar with the different (model) approaches. For example, in line 151: can you be more explicit about what you mean by 'if there is no floating ice', line 146-150 and 160-170: how does this discussion relate to quantities displayed in figure 2 (e.g. I'm unsure how 'mean ocean thermal forcing' is translated into a depth-dependent parameterization of melt), section 2.5: how is the calving front tracked in the model, and what happens to ice that has calved – is a minimum ice thickness applied?

Author response: "If there is no floating ice" means simply that the glacier margin is grounded at that cell and is not adjacent to an ice shelf. We will make this more explicit.

The thermal forcing shown in figure 2c is translated into a depth-dependent parameterization as described in Section 2.4 using the depth profile in fig 2d; however, the mean thermal forcing in 2c is only illustrative, as the full spatially and temporally varying field is used for the melt parameterization. We will make this clearer in the figure caption.

The calving front is defined as the last cell with ice (either floating or grounded) that is adjacent to ocean open ocean and has bed topography below sea level. Calving is applied as a thickness change, as MALI lacks a level-set implementation for tracking the calving front. No minimum ice thickness is applied — if the ice in a cell is completely depleted by calving or melting it simply has a thickness of 0. We will clarify this approach.

Line 274 you refer to Table 1 here, but this table does not contain any information on SLR. Also, is there a reason why \sigma_max for q=1/7 in Table 1 is different between MIROC5 and the other forcing scenarios?

Author response: Thank you, this reference to Table 1 in L274 was an oversight after changing some tables and figures. We will remove the reference to Table 1. Regarding the different value of sigma_max, this is simply an outcome of calibrating sigma_max separately for each pair of

basal friction law exponent and climate forcing. The calibration results supporting this can be found in Figure S1b. Note the different shape of the velocity misfit curves for MIROC5 vs CNRM and HadGEM2.

Line 427 I assume you are using annually averaged velocities, rather than seasonal products, for the 2007 initialization and 2017 validation? Given the large amplitude of seasonal speed-up/slow-down along this section of Greenland's margin, how important is the choice of velocity product? In lines 590-595 you allude to possible important implications, but can you provide quantitative insights in how alternative intialization approaches (e.g. by using summer-only values of surface speed) might alter/bias your results?

Author response: We use winter average velocities for the initialization and validation. To our knowledge, there are no annually averaged products available for individual years over this time period. Because it is not feasible to reproduce the seasonal velocity cycle without coupling to a validated subglacial hydrology model, we do not attempt to reproduce summer velocities. Initializing to summer velocities (if available) is possible, but would vastly overrepresent the annually-averaged ice flux from Humboldt, as the summer velocity in the north is about twice the winter velocity for a brief period. Providing a truly quantitative assessment of the impact would require another initialization of the model and set of forward runs, which would be very expensive and is well beyond the scope of this paper. We can, however, discuss this point more explicitly than we have done so far.

FigS1. Can you provide a legend for the different colours please?

Author response: Yes, thank you for pointing out this omission.

Ref.

De Rydt, J., Reese, R., Paolo, F. S., and Gudmundsson, G. H.: Drivers of Pine Island Glacier speed-up between 1996 and 2016, The Cryosphere, 15, 113–132, https://doi.org/10.5194/tc-15-113-2021, 2021
* * *
Reviewer 2:

In this work the authors investigate the contribution to sea-level rise of the Humboldt Glacier (North Greenland) for the next century. The model initial conditions are optimised through a three step procedure: first, the basal friction coefficient is optimised from surface velocity inversion at 2007; second, the basal friction exponent is tuned through imposed calving rates to match the observed ones for 2007-2017 and velocities for 2017-2018; third, the calving retreat parameterisation is tuned to match calving front positions and velocities for 2017-2018. The

resulting initialisation is then used to launch an ensemble of model simulations for the period 2007-2100 and estimate sea-level rise due to future glacier retreat.

Overall I find this a very interesting work. It is well framed, the experimental design is novel and clever, and the results are comparable to previous estimates, although higher. I think this work suits very well the scope of The Cryosphere. Yet, I am not 100% convinced about the initialisation procedure that led to such results. Since your estimated SLR contributions are considerably higher than previous estimates and you attribute this "primarily to calibration of the basal friction law to match observed surface velocity changes", I am wondering to what extent the validation procedure you apply in the optimisation+tuning experiments does affect the choice of the basal friction exponent, and so your final SLR estimates. I think that the strength of your results must be proven with some further verification of the tuning procedure for the historical runs. Moreover, I think section 2.3, as it is now, is missing some important clarifications. Therefore I suggest major revisions before publication.

Most of my comments concern the tuning of basal friction parameters in the initialisation procedure. I outline them here:

1. Could you explain better how the effective pressure N is calculated in your basal friction law (line 95)? From what you write I understand it is rho_i g H - rho_w g z_bed, right? How is N treated during the basal friction coefficient optimisation? Is it kept fixed to initial values for the whole procedure assuming that ice thickness doesn't change? See also next point.

Author Response: Effective pressure N is calculated as you have suggested where the bed is below sea-level, and is simply N = rho_i g H where the bed is above sea level. This is the most common assumption in the literature for basal friction laws that include effective pressure. During the basal friction coefficient optimization, N is calculated from the initial condition (2007 geometry), since that optimization is a snapshot inverse solution. For the tuning procedures described sections 2.3 and 2.5, as well as in all forward simulations in the ensemble and sensitivity tests, N evolves with the ice geometry.

2. The relationship used to tune the basal friction exponent (line 129, μ = μ_opt |u_opt|^(1/3-q)) should be explained more in detail. To my understanding, you derived it by solving the equation N*μ*|u_opt|^(1/3-1) = N*μ_opt*|u_opt|^(q-1), having assumed same basal friction and velocity from the inversion procedure. However, this relationship is defined under some important assumptions that should be explained. You assume that the effective pressure is the same between the optimisation and the tuning procedure, but I would expect the ice thickness has varied between 2007 and 2017 due to margin retreat, and so did N. This argument is also valid for surface velocities. How did you account for velocity changes that come out due to glacier retreat in your tuning procedure? I would expect that the choice of the best basal friction exponent ultimately depends on these assumptions. Since your results strongly depend on the value of q (Fig. 5), to what extent do you think these assumptions affect your sea-level contribution for year 2100? What happens if the relationship you wrote is not supported, i.e. the N

and velocities are not constant and, still assuming that the basal friction is the same for optimisation and tuning, you have this relationship instead: μ = N(2007) / N(2017) * μ_opt * |u_opt|^(1/3-q) * u_obs(2007)/u_mod(2017) ? Also, have you tried to do the inversion with 1/7<q<1/5 to corroborate your tuning procedure?

Author Response: The recalculation of the basal friction coefficient in L129 is derived as you have suggested, using the same basal traction, sliding speed, and effective pressure for each value of q. This is justified because this recalculation is done for the initial condition 2007 ice geometry and velocity fields; therefore, these values are the same on each side of the equation by definition. The effective pressure and velocity field at 2017 do not come into the recalculation of $\mu$. The procedure is to do this re-calculation using the initial state, after which the model is run forward to 2017 (over which time N evolves with the geometry as clarified in the previous response) and then the modeled velocities are compared to the observed at that year in order to choose which value of q is most appropriate (Fig 3). We will clarify this in the text.

We have not re-done the inversion with 1/7≤q≤1/5 because it is too computationally expensive. However, we can easily corroborate the tuning procedure by showing diagnostic velocity solutions for the 2007 initial condition using these different values of q and the one used in the optimization (q=⅓). This can be added to the supplemental material.

3. In the basal friction exponent tuning experiment you compared modelled to observed velocities only for year 2017-2018. Why didn't you test your velocities for the whole historical period (2007-2017) and choose the q that best matches the velocities on a 10yr mean? Also, would considering seasonal velocities instead of annual mean lead to a different q? Would in these cases the choice of 1/7<q<1/5 still be confirmed and so your SLR estimates?

Author Response: We calibrated the basal friction law exponent for a decade of velocity change rather than to each year in the hindcasting period to avoid calibrating our model to noisy changes in velocity and ice extent (i.e., to avoid overfitting to data). Because changes at any given time could have a number of causes (changing ice melange strength, a year of anomalously high surface or submarine melting, changing subglacial hydrology, a single large calving event, etc), calibrating to a shorter period risks lumping other processing into the basal friction law.

Reproducing seasonal velocities is a major challenge of ice-sheet modeling and would require the seasonal forcings (which are not available) and coupling the ice sheet model to a subglacial hydrology model, which is well beyond the scope of this work. In addition, seasonal velocity and ice extent observational snapshots have limited spatial and temporal coverage, which would run the risk of aliasing our results against incomplete observations.

Focusing on a single bulk tuning procedure over a decade reduces the chance of overfitting the data and is the most relevant timescale to consider for our future projections.  We agree that calibrating at annual and sub-annual timescales would be an exciting future direction, but the

observations, model, and process understanding of the system are not yet adequate to do so with confidence.

4. How did you impose the calving front retreat rates for years 2007-2018 (line 131)? To my understanding the calving tuning procedure described in section 2.5 is done after the basal friction optimisation. How did you calculate the calving rate then? Also, is the submarine melt taken into account for such tuning tests?

Author Response: You are correct that the von MIses calving calibration in section 2.5 occurs after the basal friction calibration (section 2.3), but the basal friction calibration uses an imposed decadal-mean retreat rate from observations. The imposed retreat rates are simply calculated from the difference between the observed 2007 and 2018 margin positions and the time interval in between. Retreat rate is the balance between ice flow speed and calving rate. So. to convert this retreat rate into a calving rate over the time period, we calculate the calving rate during run-time as the sum of the modeled ice velocity at each timestep and the imposed retreat rate. Submarine melting contributes to the overall retreat rate, and so is not separately active during the 10-year calibration runs for the basal friction law (Fig 3) because it is already implicitly accounted for. Submarine melting is active, however, during the 10-year calibration runs for the von Mises stress calving law (Fig S1). In this way, during optimization and calibration we explicitly separate the processes that control sliding from those that control calving.

Regarding the structure of the manuscript, I don't really understand why you separate the perturbation from the sensitivity tests. In fact, their design is comparable (you fix some parameters and perturbed some others) and they all contribute to build the uncertainty range of sea-level rise due to glacier retreat. To lighten the structure of the paper, I would suggest to include all sensitivity tests into the perturbation experiments and introduce a summary table describing the whole experimental design (which parameters are varied and which are fixed for each run). I suggest also to mark out those tests do not take part in the final estimates of sea-level contribution (e.g. tests for q=1, calving rate limit > 5km/yr). Finally, I would suggest to leave the mesh convergence test to the supplementary material, since it is more a precondition for your tests rather than a functional part of the study, and the bedrock sensitivity test too, since it does not involve any change in physical variables.

Author Response: The perturbed parameter ensemble explores the parameters that we are able to constrain based on calibrating against observations. The sensitivity tests explore parameters for which there is much deeper uncertainty: sub-shelf melt, ice-shelf collapse, calving rate limit. These parameters affect the evolution of the ice sheet and cannot be inferred using a snapshot optimization. We also include the bed topography as a sensitivity test to illustrate the effect of uncertainty in bed topography on the results, but in the absence of a Bayesian inference or Monte Carlo approach there is not a good way to include this in the ensemble. See also our response to Reviewer 1 on a similar comment.

We will take your suggestion to make it more clear which tests do not take part in the final estimate of SLR, and we can move the mesh convergence test to the supplement. We opt to keep the bed topography sensitivity test in the main text because it supports the conclusion that

uncertainty in bed topography should be taken into account in future ensembles of SLR contribution, which is a primary takeaway from this work.

Specific Comments

- Figure 1: is this the Humboldt Glacier or the regional model domain? To me that is the catchment containing the Humboldt glacier. Also, I suggest to make the black rectangle in a) with a bigger line and with a different colour. I would add the modelled effective pressure and instead of panel b) and d) I would only show the velocity difference (modelled velocity - observed velocity).

Author Response: The main figures in the panels are the regional model domain, while the insets show observations that are not on our model grid. We will add another panel showing model-data difference. Including the modeled effective pressure does not seem important, as this is one term in the equation for basal shear stress.

- Line 128: do you mean q instead of m? Also, "to find the appropriate range of values of q in the basal friction relationship, we recalculate the friction parameter ð œ‡…" is misleading. You should add you did that to match the velocities upon retreat.

Author Response: Yes, we will change this to q; m is a typo. We will add the text you suggest in the first or second sentence of this paragraph.

- Line 132: Do you mean 2017 instead of 2018? Generally, I found quite confusing the definition of the period used for hindcast, which sometimes ends in 2017, sometimes in 2018. Please check that in the whole manuscript.

Author Response: The datasets we calibrate to are provided as winter average velocities, and so 2017–2018 represents one time snapshot. We agree that this has not been communicated and that our usage is inconsistent, and we will update the text to make sure everything is clear and consistent.

- Line 156: Please change to "Connectivity Temperature Depth (CTD) and Airborne eXpendable Connectivity Temperature Depth (AXCTD)".

Author Response: We will make this change.

- Line 207: where does this SMB forcing come from? From which model? And why did you choose this period, and not a climatology close to year 2000 since you initialise the model at 2007? To what extent might the choice of a more recent climatology for the control run affect your results and reduce your estimated sea level contributions?

Author Response: This is the 1960–1989 climatology from the MIROC5 model. We use this time period to be consistent with the control runs used in the ISMIP6 ensemble to quantify model drift. We have explored using a 1995–2005 climatology from the MIROC5 model for both the q=⅕ and q=1/7 cases. As expected, using the 1995–2005 climatology leads to more sea-level

rise than the 1960–1989 climatology, but this only changes the control estimate from ~0.85 mm to ~1.5 mm SLR at 2100 and both volume change time series are linear. While using this later climatology would slightly change the magnitude of our estimates in the main ensemble, it does not change any of the interpretation.

Line 274: Table 1 does not show the results, rather summarizes the experimental design. I think that table is missing.

Yes, this text was leftover after a re-shuffle of the format. We will revise. Thanks for pointing out the oversight.

- Line 281: where does the upper bound of SLR for the 2017 Calving front experiment (6.7 mm) come from? Is HadGEM 2 predicting ~6.5 or 6.7mm?

Yes, this is from HadGEM2. We view 6.7 mm as being ~6.5 mm, but we can make this more precise.

- Line 286: with "variability due to … climate forcing" you include also the variability in submarine melting, right? Could you be more precise since the choice of the oceanic thermal forcing influences your results?

Yes, but the variability in submarine melting results directly from the variability in climate forcing. The climate forcing we refer to includes both time evolving surface mass balance and time evolving ocean thermal forcings.

- Line 304: looking at Fig. 2c it seems that CNRM-CM6 has a higher ocean thermal forcing than HadGEM2. So why does only the latter lose all the ice shelves within 2100?

This occurs because the thermal forcings are essentially normalized for the ice-shelf melt parameterization when we tune it to average 20 m/yr in the historical period. So the relevant quantity for sub-shelf melt is the change in thermal forcing over time, rather than the absolute thermal forcing. Because HadGEM2 increases so much more than CNRM, it melts a lot more. We will clarify this in the text.

- Line 321: could you introduce the undercutting already in the submarine melting parameterisation section since you have a precise parameterisation for it?

This is already introduced in the methods section, but we can ensure that the connection is explicit and the terminology is used consistently.

- Line 366: why not repeating the experiment also for MIROC for consistency with the other tests?

We chose to do this for the bounding mass loss cases, and it turns out to not make a big difference for either, so including the intermediate case (MIROC5) would not give much extra information.

Figure 9: could you plot also the change in volume above flotation and compute the associated sea-level contribution?

Yes, we will add this.

- I am missing a Figure summarising the sea-level contribution from all sensitivity/perturbation experiments compared to previous estimates. For example, you could plot the latter as superimposed to the uncertainty range in SLR raised from your runs. I think it would help the reader to have your results recapped in one plot.

Sea-level rise estimates from previous studies are not easily available, so we choose not to attempt to include these in the figure. Including sea-level contribution from the sensitivity tests would make for a very messy figure, and the purpose of those simulations was to illustrate the importance of certain modeling choices in the absence of a very large ensemble. The sensitivity tests are each one-at-a-time perturbations, so they would not fully address the range of uncertainty in SLR contribution. We will update the text to make it clear that the perturbed parameter ensemble represents our best estimate of SLR from Humboldt Glacier, while the sensitivity tests underscore several processes and properties (namely maximum calving rates, ice-shelf collapse thresholds, and bed topography) that still lend deep uncertainty to these estimates.

- Figure S3, S4: don't think you really need to show the bathymetry here. In case you want to keep it, please change the colour palette to a scale of greys. Also, specify that grounding line colours follow legend of Fig.4. Finally, please change the colour of small areas with speed>3km/yr to red or green.

We think showing bathymetry here is helpful in the context of ice dynamics, and we have chosen the color pallete to be consistent with other figures (e.g., bed topography in Fig 1 and 4). We will specify that the line colors follow the legend of Fig 4. We opt to keep the colors as they are, since we have checked these with the CoBliS colorblindness simulator and the velocity and bed topography fields are easily distinguishable. Likewise, the cyan contours around the small areas of >3 km/yr ice speed make these easily distinguishable.

---

## Author Response (AR1)

Response to reviewers' comments on "The contribution of Humboldt Glacier, North Greenland, to sea-level rise through 2100 constrained by recent observations of speedup and retreat" by Trevor R. Hillebrand, Matthew J. Hoffman, Mauro Perego, Stephen F. Price, and Ian M. Howat

We thank both reviewers and the editor for their constructive and thorough comments. Below, we reply point-by-point to the reviewer comments. Reviewer comments are in black, and author responses are in blue.

Editor: Dear Trevor Hillebrand an co-authors,

I want to sincerely thank you and your co-authors for the thorough answers and plans that you forwarded to address many of the review comments.

Both reviewers acknowledge the significance of your study to the community. Moreover, both suggest some structural changes as well as a overview chart of the experimental design.

Reviewer #1 is mostly concerned about the motivation of your 'q' calibration criterion, which combines RMSE values in three flow speed bins/classes (Fig. 3b). I wonder if some sort of objective normalisation might be helpful here (w.r.t. velocity magnitudes or logarithmic). To put it another way, does your choice for 'q' dependent on how you divide the velocity classes.

Reviewer #2 invokes the possibility to not only use the 2017/18 velocities for calibrating the sliding exponent. In terms of temporal transferability, an additional comparison with velocities prior to 2017 would be most enlightening. From Fig. 1b, you can see that there was some first acceleration between 2012-2015. You argue that other processes that might have been responsible for previous changes will thereby be lumped into the friction law exponent and you fear an overfitting. I would partially disagree. I think that your 'q' calibration ultimately depends on the fact that there was a significant speed-up of Humboldt Glacier. So before this important velocity increase, all 'q'-values might perform equally well (in terms of velocity RMSE). So your 'q'-calibration will only work in regions where we observe(d) important changes. In the best case, you can confirm the 'q' choice already in 2012-2015 and claim some transferability. In the worst case, you might need to further moderate the parameter choice in the discussion/abstract. Both outcomes would be valuable. Finally this request does not imply a lot of extra work as you only need to compute RMSE values for some further time periods.

On the basis of your point-by-point answers, I invite you to submit a revised manuscript, addressing the two major comments above: (1) normalisation of calibration criterion; (2) multi-temporal validation. In summary, I therefore suggest that your revised article will enter a second review round.

Best,

**Johannes Fürst**

We have added additional corroboration of our basal friction law exponent calibration results, and clarified the criteria and caveats behind our choice of q values. This is reflected in a nearly complete re-write of Section 2.3, as well as an overhaul of Figure 3. We have also added Figures S2–S4 to corroborate our calibration results. Figure S2 shows that our recalculation of the friction parameter using different q values successfully reproduces the initial velocity field. Figure S3 shows the 2007–2017 calibration using many narrow velocity bands in 100 m/yr increments. Figure S4 includes calibration results using the 2015–2016 observations as a target, rather than 2017–2018. The observational record prior to 2015 is too sparse to provide reliable calibration targets, and we have explained this in Section 2.3. We have also added Table 2 to summarize the sensitivity test experiments. Please refer to our detailed response to the individual reviewers, below, for more information on these major changes and other minor revisions.

**Reviewer 1:**

Authors present numerical simulations of mass loss from Humboldt Glacier. Historical runs and observations (2007-2017) are used to constrain parameters in the description of basal rheology and the calving law. Optimal model parameters are used to produce projections of mass loss for a range of future forcing scenarios (2017-2100). Results highlight the importance of the basal sliding exponent, and best estimates of mass loss exceed previous projections by about a factor of 2.

The objectives and methodology of this study are clear; the results novel and well-presented; the conclusions well-supported, and overall this work is a good fit to The Cryosphere. I recommend publications with scope for some minor revisions, as outlined below.

The experimental design is mostly straightforward and easy to follow, though I wonder about the need to distinguish between the perturbed parameter ensemble, and the additional sensitivity experiments. After all, these are all sensitivity experiments, and personally I think the distinction overcomplicates the structure of the paper. An overview of the sensitivity to all physical parameters in a single Table would be nice. Some experiments that do not test the significance of physical parameters, such as the mesh resolution and potentially, bed topo, could be included in an Appendix to reduce the amount of information in the main text.

Author response: We agree that the distinction between the perturbed parameter ensemble and the sensitivity experiments needs to be made more clear. We have added this text to the structural overview paragraph at the end of the Introduction to help readers understand the distinction: "The perturbed parameter ensemble explores the parameter space that we are able to calibrate against observations, namely basal traction and iceberg calving, and represents our best estimate of 21st century SLR from Humboldt Glacier. The sensitivity experiments are used to validate modelling choices and explore processes and characteristics that we cannot calibrate within our framework, such as submarine melt, ice-shelf collapse, maximum calving rates (which are likely much greater than anything in the observational period), and bed topography."

We have moved the mesh convergence test to the supplemental information as you suggested. The bed topography sensitivity experiment we choose to keep in the main text because the sensitivity to uncertainty in bed topography is a major conclusion of the paper that is likely applicable to many other major outlets.

**We have also added Table 2 to summarize the sensitivity tests.**

The validation approach is interesting, though I miss a more in-depth description/motivation of the validation criteria. Fig3b suggests that the optimal choice of q is critically dependent on the velocity itself, so why choose 1/5-1/7, which only provides a best match for u>600m/yr? In this regard, you might find the discussion in section 3.3 of [De Rydt et al. 2021] of interest, where authors show that the optimal sliding exponent for Pine Island Glacier is spatially heterogeneous. On a related note, I wonder if you can show the difference between observations and model in Fig3a, rather than the absolute model speed.

Author response: We chose the range of 1/5 to 1/7 not because it provided the best match to fast flow (q=1/s provides the best match in the fastest velocity band), but because it provides a reasonable match across all three velocity bands. It is true that the optimal value of the sliding exponent is very likely spatially heterogeneous due to spatially varying bed characteristics, but the majority of ice sheet models, including ours, are not currently capable of optimizing this value as a spatially varying parameter, which would require a transient optimization to invert for both mu and q. Thus, our range of 1/5–1/7 likely represents a compromise between the need for a larger exponent for slow-flowing regions and a smaller exponent for fast-flowing regions. We have added the reference to De Rydt et al. (2021) in Section 4.3: "In reality, the rheology of the bed is also likely to vary spatially, which indicates that the basal friction law exponent, q, would ideally be a spatially varying field (De Rydt et al., 2021). An improvement on our study would be to optimize q against observations as well as  $\mu$ ; however, this would require significant advances to our optimization method and is well beyond the scope of this study."

We have substantially updated the text in Section 2.3 to more clearly explain the need for the basal friction law exponent calibration and better describe its implementation. Figure 3 is also completely re-made, showing the observed 2017–2018 velocities, the model deviation from those velocities for all 9 choices of q in 2017, and normalized RMS errors for four velocity bands. We have added Figure S3, which shows that another choice of velocity band partitioning (every 100 m/yr) also suggests  $1/7 \le q \le 1/5$ . We have also added Figure S4, showing the same calibration methodology using 2015–2016 instead of 2017–2018 as the target year, which further corroborates using q in the range of 1/7-1/5.

I think some further details about the melting and calving paramterization would be instructive for readers less familiar with the different (model) approaches. For example, in line 151: can you be more explicit about what you mean by 'if there is no floating ice', line 146-150 and 160-170: how does this discussion relate to quantities displayed in figure 2 (e.g. I'm unsure how 'mean ocean thermal forcing' is translated into a depth-dependent parameterization of melt), section 2.5: how is the calving front tracked in the model, and what happens to ice that has calved – is a minimum ice thickness applied?

**Author response:**

In Section 2.4, we have clarified that there are two separate parameterizations used for submarine melt. We removed the phrase "if there is no floating ice", which proved to be redundant as we are explicitly discussing the grounded margin in this paragraph.

The thermal forcing shown in figure 2c is translated into a depth-dependent parameterization as described in Section 2.4 using the depth profile in fig 2d; however, the mean thermal forcing in 2c is only illustrative, as the full spatially and temporally varying field is used for the melt parameterization. We have added this text to the caption of Figure 2 to clarify this: "The total and mean values are depicted here for illustration, but the full time-varying, two-dimensional fields are used to force the simulations."

We have added this text to Section 2.5 to clarify the calving implementation: "MALI lacks a sub-grid tracking scheme for the calving front; instead, the calving front is tracked as described by Hoffman et al. (2018), following the method of Albrecht et al. (2011) that includes a row of thin non-dynamic cells at the marine margin to conserve mass. The marginal dynamic cells are considered to be the calving front and their stress and velocity values are used to calculate the local calving velocity. Calved ice is instantaneously removed from the domain."

**Line 274 you refer to Table 1 here, but this table does not contain any information on SLR. Also, is there a reason why \sigma\_max for q=1/7 in Table 1 is different between MIROC5 and the other forcing scenarios?**

Author response: Thank you, this reference to Table 1 in L274 was an oversight after changing some tables and figures. We removed the reference to Table 1. Regarding the different value of sigma\_max, this is simply an outcome of calibrating sigma\_max separately for each pair of basal friction law exponent and climate forcing. We have added a reference in the table caption to Figure S5, which shows the calibration results.

Line 427 I assume you are using annually averaged velocities, rather than seasonal products, for the 2007 initialization and 2017 validation? Given the large amplitude of seasonal speed-up/slow-down along this section of Greenland's margin, how important is the choice of velocity product? In lines 590-595 you allude to possible important implications, but can you provide quantitative insights in how alternative initialization approaches (e.g. by using summer-only values of surface speed) might alter/bias your results?

Author response: We use winter-average velocities for the initialization and validation. To our knowledge, there are no annually averaged products available for individual years over this time period. Because it is not feasible to reproduce the seasonal velocity cycle without coupling to a validated subglacial hydrology model, we do not attempt to reproduce summer velocities. Initializing to summer velocities (if available) is possible, but would vastly overrepresent the annually-averaged ice flux from Humboldt, as the summer velocity in the north is about twice the winter velocity for a brief period. Providing a truly quantitative assessment of the impact would require another initialization of the model and set of forward runs, which would be very expensive and is well beyond the scope of this paper.

We have clarified in Sections 2.2, 2.3 and 4.1 that we use winter velocities for the optimization and calibration. We have also added the following text to Section 4.1 to discuss how using summer or mean-annual products (if they were readily available) would likely impact our results: "If the seasonal cycle persists through the 21st century, our simulated mass loss could be an underestimate, as we do not capture the summer months during which flux through the glacier is much larger than the annual average. However, using mean-annual velocities instead of winter velocities would impose a modeled glacier state that only represents reality for a very small fraction of the year, while using only summer velocities would greatly overestimate dynamic mass loss. Thus, in the absence of a modeled seasonal velocity cycle, using winter velocities for initialization and calibration provides a more conservative but also more representative initial model state compared with using mean annual or summer velocities."

**FigS1. Can you provide a legend for the different colours please?**

Author response: Yes, thank you for pointing out this omission. We have added the legend. This is now Figure S4.

Ref.

De Rydt, J., Reese, R., Paolo, F. S., and Gudmundsson, G. H.: Drivers of Pine Island Glacier speed-up between 1996 and 2016, The Cryosphere, 15, 113–132, https://doi.org/10.5194/tc-15-113-2021, 2021

Reviewer 2:

In this work the authors investigate the contribution to sea-level rise of the Humboldt Glacier (North Greenland) for the next century. The model initial conditions are optimised through a three step procedure: first, the basal friction coefficient is optimised from surface velocity

inversion at 2007; second, the basal friction exponent is tuned through imposed calving rates to match the observed ones for 2007-2017 and velocities for 2017-2018; third, the calving retreat parameterisation is tuned to match calving front positions and velocities for 2017-2018. The resulting initialisation is then used to launch an ensemble of model simulations for the period 2007-2100 and estimate sea-level rise due to future glacier retreat.

Overall I find this a very interesting work. It is well framed, the experimental design is novel and clever, and the results are comparable to previous estimates, although higher. I think this work suits very well the scope of The Cryosphere. Yet, I am not 100% convinced about the initialisation procedure that led to such results. Since your estimated SLR contributions are considerably higher than previous estimates and you attribute this "primarily to calibration of the basal friction law to match observed surface velocity changes", I am wondering to what extent the validation procedure you apply in the optimisation+tuning experiments does affect the choice of the basal friction exponent, and so your final SLR estimates. I think that the strength of your results must be proven with some further verification of the tuning procedure for the historical runs. Moreover, I think section 2.3, as it is now, is missing some important clarifications. Therefore I suggest major revisions before publication.

Most of my comments concern the tuning of basal friction parameters in the initialisation procedure. I outline them here:

 Could you explain better how the effective pressure N is calculated in your basal friction law (line 95)? From what you write I understand it is rho\_i g H - rho\_w g z\_bed, right? How is N treated during the basal friction coefficient optimisation? Is it kept fixed to initial values for the whole procedure assuming that ice thickness doesn't change? See also next point.

Author Response: Effective pressure N is calculated as you have suggested where the bed is below sea-level, and is simply N = rho\_i g H where the bed is above sea level. This is the most common assumption in the literature for basal friction laws that include effective pressure. During the basal friction coefficient optimization, N is calculated from the initial condition (2007 geometry), since that optimization is a snapshot inverse solution. For the tuning procedures described sections 2.3 and 2.5, as well as in all forward simulations in the ensemble and sensitivity tests, N evolves with the ice geometry. We have clarified this in section 2.1: "N evolves based on ice geometry in all simulations."

2. The relationship used to tune the basal friction exponent (line 129, μ = μ\_opt |u\_opt|^(1/3-q)) should be explained more in detail. To my understanding, you derived it by solving the equation N\*μ\*|u\_opt|^(1/3-1) = N\*μ\_opt\*|u\_opt|^(q-1), having assumed same basal friction and velocity from the inversion procedure. However, this relationship is defined under some important assumptions that should be explained. You assume that the effective pressure is the same between the optimisation and the tuning procedure, but I would expect the ice thickness has varied between 2007 and 2017 due to margin retreat, and so did N. This argument is also valid for surface velocities. How did you account for velocity changes that come out due to glacier retreat in your tuning

procedure? I would expect that the choice of the best basal friction exponent ultimately depends on these assumptions. Since your results strongly depend on the value of q (Fig. 5), to what extent do you think these assumptions affect your sea-level contribution for year 2100? What happens if the relationship you wrote is not supported, i.e. the N and velocities are not constant and, still assuming that the basal friction is the same for optimisation and tuning, you have this relationship instead:  $\mu = N(2007) / N(2017) * \mu_opt * |u_opt|^(1/3-q) * u_obs(2007)/u_mod(2017) ? Also, have you tried to do the inversion with 1/7<q<1/5 to corroborate your tuning procedure?$

Author Response: The recalculation of the basal friction coefficient is derived using the same basal traction, sliding speed, and effective pressure for each value of g. This is justified because this recalculation is done for the initial condition 2007 ice geometry and velocity fields; therefore, these values are the same on each side of the equation by definition. The effective pressure and velocity field at 2017 do not come into the recalculation of  $\mu$ . The procedure is to do this re-calculation using the initial state, after which the model is run forward to 2017 (over which time N evolves with the geometry as clarified in the previous response) and then the modeled velocities are compared to the observed at that year in order to choose which value of q is most appropriate (Fig 3). We have substantially revised Section 2.3 and Figure 3 to make this much more clear. We have not re-done the inversion with  $1/7 \le q \le 1/5$  because it is too computationally expensive. However, we can easily corroborate the tuning procedure by showing diagnostic velocity solutions for the 2007 initial condition using these different values of q and the one used in the optimization  $(q=\frac{1}{3})$ . We have also added Figure S2 to show that using  $q=\frac{1}{3}$  and 1/7 gives us very nearly the same initial velocity field as the optimization, which validates our recalculation. Figures S3 and S4 further corroborate our choice of g (see response to reviewer 1 on similar point).

3. In the basal friction exponent tuning experiment you compared modelled to observed velocities only for year 2017-2018. Why didn't you test your velocities for the whole historical period (2007-2017) and choose the q that best matches the velocities on a 10yr mean? Also, would considering seasonal velocities instead of annual mean lead to a different q? Would in these cases the choice of 1/7<q<1/5 still be confirmed and so your SLR estimates?

Author Response: We calibrated the basal friction law exponent for a decade of velocity change rather than to each year in the hindcasting period to avoid calibrating our model to noisy changes in velocity and ice extent (i.e., to avoid overfitting to data). Because changes at any given time could have a number of causes (changing ice melange strength, a year of anomalously high surface or submarine melting, changing subglacial hydrology, a single large calving event, etc), calibrating to a shorter period risks lumping other processing into the basal friction law.

Reproducing seasonal velocities is a major challenge of ice-sheet modeling and would require the seasonal forcings (which are not available) and coupling the ice sheet model to a subglacial hydrology model, which is well beyond the scope of this work. In addition, seasonal velocity and ice extent observational snapshots have limited spatial and temporal coverage, which would run the risk of aliasing our results against incomplete observations.

Focusing on a single bulk tuning procedure over a decade reduces the chance of overfitting the data and is the most relevant timescale to consider for our future projections. We agree that calibrating at annual and sub-annual timescales would be an exciting future direction, but the observations, model, and process understanding of the system are not yet adequate to do so with confidence.

Based on this comment, we have better justified our choice of the 2007–2008 to 2017–2018 changes in section 2.3: "We perform the calibration over a full decade of change to take advantage of the largest signal of velocity change available. This also avoids calibrating to noisy interannual variability and aliasing to temporal gaps in the data; for instance, the data gap between 2008–2009 and 2012–2013 (Figure 1) means that we cannot say whether the glacier retreated monotonically during this period, or if it retreated and then readvanced to the 2012 margin."

We have also added Figure S3 to test our calibration method over the 2007–2008 to 20015–2016 period, which confirms our choice of q values.

4. How did you impose the calving front retreat rates for years 2007-2018 (line 131)? To my understanding the calving tuning procedure described in section 2.5 is done after the basal friction optimisation. How did you calculate the calving rate then? Also, is the submarine melt taken into account for such tuning tests?

Author Response: You are correct that the von MIses calving calibration in section 2.5 occurs after the basal friction calibration (section 2.3), but the basal friction calibration uses an imposed decadal-mean retreat rate from observations. The imposed retreat rates are simply calculated from the difference between the observed 2007 and 2018 margin positions and the time interval in between. Retreat rate is the balance between ice flow speed and calving rate. So to convert this retreat rate into a calving rate over the time period, we calculate the calving rate during run-time as the sum of the modeled ice velocity at each timestep and the imposed retreat rate. We have clarified that the basal friction law exponent calibration step comes before the von Mises stress calving calibration step at the end of section 1: "We first calibrate the basal friction law exponent using an imposed retreat rate. We then tune an iceberg calving parameterization using the range of calibrated basal friction law exponents." We reiterate this for clarity in section 2.5: "To tune the von Mises tensile threshold stress parameter,  $\sigma_{max}$ , we ran the model from winter 2007–2008 to winter 2017–2018 with values of  $\sigma_{max}$  of 150, 160, 170, 180, 190, and 200 kPa for all pairs of climate forcing and calibrated basal friction law exponent (q = 1/5 and 1/7) "

Submarine melting contributes to the overall retreat rate, and so is not separately active during the 10-year calibration runs for the basal friction law (Fig 3) because it is already implicitly accounted for. We have clarified this in 2.3: "Aside from the imposed calving rate, the only other forcing in this step is the MIROC5 RCP8.5 surface mass balance. The imposed calving rate

represents the sum of retreat due to calving and submarine melting in reality, the individual components of which are not readily available from observations. "

Submarine melting is active, however, during the 10-year calibration runs for the von Mises stress calving law (Fig S4). We have added this text to 2.5: "Submarine melting is active during the von Mises calving calibration runs."

**In this way, during optimization and calibration we explicitly separate the processes that control sliding from those that control calving.**

Regarding the structure of the manuscript, I don't really understand why you separate the perturbation from the sensitivity tests. In fact, their design is comparable (you fix some parameters and perturbed some others) and they all contribute to build the uncertainty range of sea-level rise due to glacier retreat. To lighten the structure of the paper, I would suggest to include all sensitivity tests into the perturbation experiments and introduce a summary table describing the whole experimental design (which parameters are varied and which are fixed for each run). I suggest also to mark out those tests do not take part in the final estimates of sea-level contribution (e.g. tests for q=1, calving rate limit > 5km/yr). Finally, I would suggest to leave the mesh convergence test to the supplementary material, since it is more a precondition for your tests rather than a functional part of the study, and the bedrock sensitivity test too, since it does not involve any change in physical variables.

Author Response: The perturbed parameter ensemble explores the parameters that we are able to constrain based on calibrating against observations. The sensitivity tests explore parameters for which there is much deeper uncertainty: sub-shelf melt, ice-shelf collapse, calving rate limit. These parameters affect the evolution of the ice sheet and cannot be inferred using a snapshot optimization. We also include the bed topography as a sensitivity test to illustrate the effect of uncertainty in bed topography on the results, but in the absence of a Bayesian inference/Monte Carlo approach there is not a good way to include this in the ensemble. It is not feasible to integrate the sensitivity tests into the perturbed parameter ensemble because not all combinations are explored, and because the point of the sensitivity tests is to validate some configurations and invalidate others. For example, it would not make sense to include the calving rate limit tests from 3.3.3 in the full ensemble because only a subsection of those runs result in scientifically valid behavior. Likewise, ice-shelf collapse (3.3.1) is an extremely uncertain process, and including it in our perturbed parameter ensemble would double the ensemble size without adding any further insight. We have added Table 2 to summarize the sensitivity experiments, which also helps distinguish between the perturbed parameter ensemble (Table 1) and sensitivity tests.

See also our response to Reviewer 1 on a similar comment. We have clarified this distinction at the end of Section 1: "The perturbed parameter ensemble explores the parameter space that we are able to calibrate against observations, namely basal traction and iceberg calving, and represents our best estimate of 21st century SLR from Humboldt Glacier. The sensitivity experiments encompass processes and characteristics that we cannot calibrate within our framework, such as submarine melt, ice-shelf collapse, maximum calving rates (which are likely much greater than anything in the observational

period), and bed topography." There is not an obvious way to mark out the tests that do not take part in the final estimate of sea-level contribution in the figures, so we have made sure the text clearly states that the perturbed parameter ensemble is our best estimate of SLR.

We have moved the mesh convergence test to the supplement.

We opt to keep the bed topography sensitivity test in the main text because it supports the conclusion that uncertainty in bed topography should be taken into account in future ensembles of SLR contribution, which is a primary takeaway from this work.

**Specific Comments**

• Figure 1: is this the Humboldt Glacier or the regional model domain? To me that is the catchment containing the Humboldt glacier. Also, I suggest to make the black rectangle in a) with a bigger line and with a different colour. I would add the modelled effective pressure and instead of panel b) and d) I would only show the velocity difference (modelled velocity - observed velocity).

Author Response: The main figures in the panels are the regional model domain, while the insets show observations that are not on our model grid. We have made the rectangle in panel (a) thicker, but choose to keep it black because there are so many colors in this figure that a different color would risk confusing the meaning; we think the thicker line makes it visible enough. Showing the observed velocity is much more important than showing the modeled effective pressure, as effective pressure is just one term in the equation for basal shear stress and follows a very commonly used parameterization. We have added the absolute and relative surface velocity errors for the optimization in Figure S1.

Line 128: do you mean q instead of m? Also, "to find the appropriate range of values of q in the basal friction relationship, we recalculate the friction parameter ð at..." is misleading. You should add you did that to match the velocities upon retreat.

**Author Response: Yes, changed to q; m was a typo.**

We have added this text to the first paragraph of 2.3, which addresses your concern in the context of the heavily revised section: "Our optimized initial condition provides basal shear stress and velocity for the initial condition, but the evolution of these fields depends on the value of the basal friction relationship exponent, *q*, which is not known *a priori* and cannot be determined from a single snapshot in time (Gillet-Chaulet et al., 2016; Joughin et al., 2019). Thus, to find the appropriate range of values of *q*, we recalculate the friction parameter  $\mu$  for  $1/10 \le q \le 1$  and evaluate each case against observed velocities after a decade of forward integration. "

• Line 132: Do you mean 2017 instead of 2018? Generally, I found quite confusing the definition of the period used for hindcast, which sometimes ends in 2017, sometimes in 2018. Please check that in the whole manuscript.

Author Response: The datasets we calibrate to are provided as winter average velocities, and so 2017–2018 represents one time snapshot. We agree that this has not been communicated and that our usage is inconsistent, and we have updated the text accordingly.

• Line 156: Please change to "Connectivity Temperature Depth (CTD) and Airborne eXpendable Connectivity Temperature Depth (AXCTD)".

Author Response: We have added "Conductivity, Temperature, Depth (CTD)" and "Airborne eXpendable Conductivity, Temperature, Depth (AXCTD)"

• Line 207: where does this SMB forcing come from? From which model? And why did you choose this period, and not a climatology close to year 2000 since you initialise the model at 2007? To what extent might the choice of a more recent climatology for the control run affect your results and reduce your estimated sea level contributions?

Author Response: We were previously using the 1960–1989 climatology from the MIROC5 model for all cases, but this comment led us to run separate control runs for each climate forcing-basal friction law pair (i.e., six in all for the perturbed parameter ensemble). We use this time period to be consistent with the control runs used in the ISMIP6 ensemble to quantify model drift. We have explored using a 1995–2005 climatology from the MIROC5 model for both the q=1/2 and q=1/7 cases. As expected, using the 1995–2005 climatology leads to more sea-level rise than the 1960–1989 climatology; this changes the control estimate from ~0.85 mm to ~1.5 mm SLR at 2100 and both volume change time series are close to linear. While using this later climatology would slightly change the magnitude of our estimates in the main ensemble, it does not change any of the interpretation. However, in order to account for drift in our model under a climate that was not causing rapid retreat of the ice sheet, we think using the 1960-1989 climatology is a better choice. We have updated section 2.6.1 to better explain our control simulations. We have also updated Figures 4-5 and 7-10 to use the new, larger set of control simulations. This brings down the upper bound of SLR by about 0.5 mm, but does not change any major interpretations. We have also updated reported SLR throughout the text accordingly.

Line 274: Table 1 does not show the results, rather summarizes the experimental design. I think that table is missing.

Yes, this text was leftover after a re-shuffle of the format. We have revised this. Thanks for pointing out the oversight.

• Line 281: where does the upper bound of SLR for the 2017 Calving front experiment (6.7 mm) come from? Is HadGEM 2 predicting ~6.5 or 6.7mm?

Yes, this is from HadGEM2. We view 6.7 mm as being ~6.5 mm, but we have made this more precise. Note that with the new control runs subtracted off, the range is now 5.8–6.1.

• Line 286: with "variability due to ... climate forcing" you include also the variability in submarine melting, right? Could you be more precise since the choice of the oceanic thermal forcing influences your results?

Yes, but the variability in submarine melting results directly from the variability in climate forcing. The climate forcing we refer to includes both time evolving surface mass balance and time evolving ocean thermal forcings. We have clarified this in this sentence by changing "climate forcing" to "atmospheric and oceanic forcing"

• Line 304: looking at Fig. 2c it seems that CNRM-CM6 has a higher ocean thermal forcing than HadGEM2. So why does only the latter lose all the ice shelves within 2100?

This occurs because the thermal forcings are essentially normalized for the ice-shelf melt parameterization when we tune it to average 20 m/yr in the historical period. So the relevant quantity for sub-shelf melt is the change in thermal forcing over time, rather than the absolute thermal forcing. Because HadGEM2 increases so much more than CNRM, it melts a lot more. We have clarified this in 2.4: "This tuning effectively normalizes the thermal forcing fields shown in Figure 2, so that the relevant quantity for sub-shelf melt is the change in thermal forcing relative to 2007, while the relevant quantity for undercutting is the absolute value of thermal forcing. "

• Line 321: could you introduce the undercutting already in the submarine melting parameterisation section since you have a precise parameterisation for it?

This is already introduced in the methods section, but have changed the wording here and in the submarine melting parameterizations section (2.4) to ensure consistent terminology.

• Line 366: why not repeating the experiment also for MIROC for consistency with the other tests?

We chose to do this for the bounding mass loss cases, and it turns out to not make a significant difference for either, so including the intermediate case (MIROC5) would not give much extra information.

Figure 9: could you plot also the change in volume above flotation and compute the associated sea-level contribution?

Yes, we have added this.

• I am missing a Figure summarising the sea-level contribution from all sensitivity/perturbation experiments compared to previous estimates. For example, you could plot the latter as superimposed to the uncertainty range in SLR raised from your runs. I think it would help the reader to have your results recapped in one plot.

Sea-level rise estimates from previous studies are not easily available, so we choose not to attempt to include these in the figure. Including sea-level contribution from the sensitivity tests would make for a very messy figure, and the purpose of the sensitivity simulations was to illustrate the importance of certain modeling choices in the absence of a very large ensemble, rather than to make true SLR estimates. The sensitivity tests are each one-at-a-time perturbations, so they would not fully address the range of uncertainty in SLR contribution. We have updated the end of Section 1 to make it clear that the perturbed parameter ensemble represents our best estimate of SLR from Humboldt Glacier, while the sensitivity tests underscore several processes and properties (namely maximum calving rates, ice-shelf collapse thresholds, and bed topography) that still lend deep uncertainty to these estimates: "The perturbed parameter ensemble explores the parameter space that we are able to calibrate against observations, namely basal traction and iceberg calving, and represents our best estimate of 21st century SLR from Humboldt Glacier. The sensitivity experiments are used to validate modelling choices and explore processes and characteristics that we cannot calibrate within our framework, such as submarine melt, ice-shelf collapse, maximum calving rates (which are likely much greater than anything in the observational period), and bed topography. "

• Figure S3, S4: don't think you really need to show the bathymetry here. In case you want to keep it, please change the colour palette to a scale of greys. Also, specify that grounding line colours follow legend of Fig.4. Finally, please change the colour of small areas with speed>3km/yr to red or green.

We think showing bathymetry here is helpful in the context of ice dynamics, and we have chosen the color pallete to be consistent with bed topography in Figures 1 and 4 in the main text; a grey color palette would be very difficult, since we need colors that diverge about sea level. We opt to keep the colors as they are, since we have checked these with the CoBliS colorblindness simulator (https://www.color-blindness.com/coblis-color-blindness-simulator/) and the velocity and bed topography fields are easily distinguishable. Likewise, the cyan contours around the small areas of >3 km/yr ice speed make these easily distinguishable and are not easily confused with the grounding line contour colors. We have added a note that grounding line colors follow the legend of Figure 4.

---

## Author Response (AR2)

We thank the reviewer for these helpful comments, which have further improved details of the manuscript. Please find our responses to their comments below.

Reviewer comment: Section 2.4: I would still like to see the equation for basal melt as a function of TF here. Also, consider adding some more details on how the basal melt parameterization is tuned to reproduce 20 m/yr average melt. What physical parameters in the parameterization are you tuning, and are these tuned values within reasonable bounds?

Author response: We have added equation 2 to show how we calculate melt based on thermal forcing, and explained that we tune a parameter analogous to exchange rate, following Jourdain et al. (2020). We have added this text to clarify the parameter tuning: "We find  $\gamma$ 0 values of 30,246 m yr-1 for CNRM-CM6, 45,292 m yr-1 for HadGEM2, and 75,580 m yr-1 for MIROC5. For reference, Jourdain et al. (2020) found median  $\gamma$ 0 values for Antarctica of 14,500 and 159,000 m yr-1 for their two calibration methods."

Reviewer comment: Supplementary Figure S8a: The velocity maps do not extent to the 2017 ice front location (as in Fig S7a), which I interpret as the ice front being retreated inland of its 2017 position. Is that because ice thickness has reduced to zero here?

Author response: That's correct. The ice front has retreated due to combined surface and submarine melting. We have clarified this in the figure caption: "The retreat in simulations using the 2017 calving front (a) is due combined surface and sub-marine melting."

Reviewer comment: L441-451 From Fig2 it looks like CNRM-CM6 exhibits a similar rise in ocean thermal forcing toward the end of the century, yet the ice shelves remain intact. Is there an obvious explanation?

Author response: Thanks for pointing this out! We were naively showing thermal forcing averaged over the whole marine-based part of the domain, but much of that is not actually experienced by the glacier during our simulations. HadGEM2 warms much more in the deep central-to-northern part of the fjord than CNRM does. We have changed this figure to show the average of the thermal forcing between the initial ice extent (2007) and the most retreated configuration of the grounding line in the perturbed parameter ensemble (HadGEM2, q=1/7, low sigma max at 2100), which now shows much more clearly that HadGEM2 warms significantly more in the relevant area. Also note that the absolute values are much larger when averaged in this way, but this does not change the interpretation. We have added the explanatory text to the caption: "The values in (c) were calculated using the area between the initial ice front and the most retreated grounding-line position in our perturbed parameter ensemble, where the bed is below sea level."

Reviewer comment: L725 and following: you might also find the recently accepted results by Barnes and Gudmundsson (2022) of interest. They broadly support your discussion about the complex dependency of glacier response on different sliding parameterizations.

Barnes, J. M. and Gudmundsson, G. H.: The predictive power of ice sheet models and the regional sensitivity of ice loss to basal sliding parameterisations: A case study of Pine Island and

Thwaites Glaciers, West Antarctica, The Cryosphere Discuss. [preprint], https://doi.org/10.5194/tc-2022-109, in review, 2022.

Author response: We have added this reference to the sentence: "However, there does not seem to be any consensus on the best form of the basal friction relationship, and the best choice could be glacier- or basin-dependent (Barnes & Gudmundsson, 2022)."